# Marine ice-sheet experiments with the Community Ice Sheet Model

Gunter R. Leguy[1], William H. Lipscomb[1], and Xylar S. Asay-Davis[2]

[1]Climate and Global Dynamics Laboratory, National Center for Atmospheric Research, Boulder, CO, USA
[2]Los Alamos National Laboratory, Los Alamos, New Mexico, USA

**Correspondence:** G. R. Leguy (gunterl@ucar.edu)

**Abstract.**

Ice sheet models differ in their numerical treatment of dynamical processes. Simulations of marine-based ice are sensitive to the choice of Stokes flow approximation and basal friction law, and to the treatment of stresses and melt rates near the grounding line. We study the effects of these numerical choices on marine ice-sheet dynamics in the Community Ice Sheet Model (CISM). In the framework of the Marine Ice Sheet Model Intercomparison Project 3d (MISMIP3d), we show that a depth-integrated, higher-order solver gives results similar to a 3D (Blatter-Pattyn) solver. We confirm that using a grounding-line parameterization to approximate stresses in the grounding zone leads to accurate representation of ice sheet flow with a resolution of ~2 km, as opposed to ~0.5 km without the parameterization. In the MISMIP+ experimental framework, we compare different treatments of sub-shelf melting near the grounding line. In contrast to recent studies arguing that melting should not be applied in partly grounded cells, it is usually beneficial in CISM simulations to apply some melting in these cells. This suggests that the optimal treatment of melting near the grounding line can depend on ice-sheet geometry, forcing, or model numerics. In both experimental frameworks, ice flow is sensitive to the choice of basal friction law. To study this sensitivity, we evaluate friction laws that vary the connectivity between the basal hydrological system and the ocean near the grounding line. CISM yields accurate results in steady-state and perturbation experiments at a resolution of ~2 km (arguably 4 km) when the connectivity is low or moderate, and ~1 km (arguably 2 km) when the connectivity is strong.

## 1 Introduction

The Antarctic Ice Sheet is losing mass (Shepherd et al., 2018; Rignot et al., 2019), primarily from marine-based parts of the West Antarctic Ice Sheet (WAIS). Warming in the Southern Ocean and increased penetration of warm water into ice-shelf cavities have led to increased basal melting and ice-shelf thinning (Hellmer et al., 2012; Pritchard et al., 2012; Sallée, 2018). Consequently, the floating ice offers less resistance and allows acceleration and increased discharge of grounded ice, leading to retreat of the grounding line where the ice lifts off the bed and becomes afloat. In regions where the seabed deepens upstream of the grounding line, this retreat can be self-reinforcing (Schoof, 2007a). Research is ongoing (Seroussi et al., 2020; Rückamp et al., 2020; Lipscomb et al., 2021a) to better understand and model the physical processes responsible for marine ice-sheet behavior, in order to better forecast future mass loss and sea-level rise.

Several equation sets of varying complexity have been derived to represent ice sheet flow. The most complex models to date are those that solve the Stokes equations (Durand et al., 2009; Favier et al., 2012; Leng et al., 2012; Tezaur et al., 2015).

These are the most accurate models, as they represent all components of the stress tensor, but they are expensive to run at continental scale. Many ice sheet models reduce computational cost and complexity by solving various Stokes approximations (Pattyn et al., 2008; Schoof and Hindmarsh, 2010), neglecting terms in the stress tensor that are less important for a given application. These include the three-dimensional higher-order approximation developed by Blatter (1995) and Pattyn (2003), along with depth-integrated higher-order approximations (Goldberg, 2011; Perego et al., 2012). Other models solve the simpler shallow-shelf approximation (SSA; MacAyeal, 1989; Schoof, 2007a) or shallow-ice approximation (SIA; Hutter, 1983). There are also hybrid models that combine Stokes and the SIA (Ahlkrona et al., 2016), or the SIA and SSA (Bueler and Brown, 2009; Martin et al., 2011; Pollard and DeConto, 2012). The SIA is valid only for strongly grounded ice in which vertical shear stresses are dominant. The other approximations include lateral and longitudinal stresses and are more appropriate for modeling marine-based ice, in which vertical shear can be small or zero.

For marine ice sheets, a particular challenge is to model dynamical processes in the grounding zone, where inadequate grid resolution can lead to inaccurate results (Vieli and Payne, 2005; Leguy et al., 2014; Gladstone et al., 2017; Seroussi and Morlighem, 2018). The representation of basal friction near the grounding line greatly influences the dynamic response. The Marine Ice Sheet Model Intercomparison Project (MISMIP; Pattyn et al., 2012) showed that one-dimensional models with discontinuous basal friction across the grounding line (Schoof, 2007b) require a resolution of ∼100 m to accurately represent grounding-line motion. At this resolution it can be computationally prohibitive to simulate an entire ice sheet, motivating modelers to find cheaper ways to simulate grounding lines.

In higher-order models, the required resolution is coarser for sliding laws in which basal stresses are continuous across the grounding line (Leguy et al., 2014; Tsai et al., 2015; Gladstone et al., 2017) (This is not necessarily true for full-Stokes models, as discussed by Gagliardini et al. 2016). Using the idealized MISMIP experiments, Leguy et al. (2014) showed that when basal friction decreases smoothly to zero, grounding-line motion can be represented accurately at ∼1 km resolution. Such a transition is physically possible when the hydrological system below the grounded ice sheet is connected to the ocean, and the water pressure equals the ice overburden pressure at the grounding line.

Resolution requirements can also be reduced with grounding-line parameterizations (GLPs), which represent subgrid-scale variations in basal friction. Typically, GLPs reduce the basal shear stress in grid cells at the grounding line in proportion to the fraction of the cell that is floating. With a GLP, a resolution of 1–2 km may be sufficient to accurately represent grounding-line motion (Gladstone et al., 2010; Seroussi et al., 2014; Leguy et al., 2014; Cornford et al., 2016), with either continuous or discontinuous basal friction. The required resolution can depend on several factors, including basal drag, channel width, and bed topography (Gladstone et al., 2012). GLPs are beneficial for Stokes-approximation solvers, as well as models configured to solve the full Stokes flow equations (Cheng et al., 2020). GLPs have been implemented in several three-dimensional models to reduce the computational cost of Antarctic simulations (Seroussi et al., 2014; Cornford et al., 2016; Lipscomb et al., 2021a).

Adaptive mesh refinement (AMR), in which the grid evolves during the simulation, also can be used to represent grounding lines more accurately (Cornford et al., 2013). This method has the advantage of using high resolution only where needed, with coarser resolution elsewhere, at the cost of additional model complexity. Cornford et al. (2016) found that adding a GLP to

AMR allows simulations to be run on a grid 2 to 4 times coarser than without a GLP. With a GLP, a resolution of ∼1 km near grounding lines can be appropriate for full Antarctic simulations.

The application of basal melting beneath floating ice is also an active research topic (Asay-Davis et al., 2017), playing a major role in simulating ice–ocean interactions (Asay-Davis et al., 2016, 2017; Favier et al., 2019; Seroussi et al., 2020).
Recent studies (Seroussi and Morlighem, 2018; Cornford et al., 2020) have suggested that, to reduce numerical errors, no melting should be applied in the grid cell containing the grounding line. A physical argument can be made that applying melt in the cell containing the grounding line will artificially drive retreat, by thinning grounded ice upstream of the grounding line. We will present results claiming otherwise: that the optimal treatment of basal melt near the grounding line may depend on the physical situation and the details of a specific model, and could include some melting in partly grounded cells.
In this paper, we extend the work of Leguy et al. (2014) to 3D models using the Community Ice Sheet Model (CISM; Lipscomb et al., 2019). CISM was first developed for applications to land-based ice sheets, especially Greenland (Lipscomb et al., 2013; Goelzer et al., 2018). CISM has been coupled to the Community Earth System Model (CESM; Hurrell et al., 2013; Danabasoglu et al., 2020) and used in interactive ice sheet–climate simulations with a dynamic Greenland Ice Sheet (Muntjewerf et al., 2020a, b). Recently, CISM has been used for standalone Antarctic Ice Sheet simulations (Seroussi et al.,
2019, 2020; Levermann et al., 2019; Lipscomb et al., 2021a), and work is underway to extend CESM–CISM coupling to Antarctica and other marine-based ice sheets. Antarctic applications have motivated us to test the accuracy and robustness of model numerics related to grounding-line dynamics. To this end, we will use the MISMIP3d (Pattyn et al., 2013) and MISMIP+ (Asay-Davis et al., 2016; Cornford et al., 2020) experiments as a framework to show that the conclusions of Leguy et al. (2014) remain valid: grid resolution of ∼1 km (and, in some circumstances, coarser) is sufficient to accurately capture grounding-line
migration when using a GLP, when the ice sheet is hydrologically connected to the ocean, or both. We extend that study to consider the treatment of basal melting near the grounding line and the choice of Stokes approximation.

Section 2 describes the CISM developments used to perform the experiments in this study. In Sects. 3 and 4, we present simulation results using the MISMIP3d and MISMIP+ experimental frameworks, respectively. We discuss the results and conclude in Sect. 5.

## 2   Model description

We use a developmental version of CISM, building on v2.1, which includes several approximations of Stokes flow. In this study we compare results using the shallow-shelf approximation (SSA), a Depth-Integrated Viscosity Approximation (DIVA, based on Goldberg 2011), and the Blatter-Pattyn (BP) approximation. Lipscomb et al. (2019) describe the solution methods for each velocity solver, along with other features of CISM's dynamical core. This section focuses on the parameterizations used
in CISM to treat basal friction and sub-ice-shelf melting near the grounding line of marine ice sheets.

The CISM grid is shown schematically in Fig. 1. Scalars such as ice thickness $H$ and bed topography $b$ are located at grid cell centers, with ice velocity components $(u, v)$ at cell vertices. Since basal melt rates modify $H$, they lie at cell centers, whereas

basal friction is a forcing term for velocity and is defined at vertices. This staggering of variables is incorporated in the GLPs for friction and sub-shelf melting, as discussed below.

## 2.1 Basal sliding law

We use the same basal sliding law as in Leguy et al. (2014), based on Schoof (2005):

$$\boldsymbol{\tau_b} = C|\boldsymbol{u}|^{\frac{1}{n}-1}\left(\frac{N^n}{\kappa|\boldsymbol{u}|+N^n}\right)^{\frac{1}{n}}\boldsymbol{u}, \tag{1}$$

where $\boldsymbol{\tau_b}$ is the basal shear stress, $\boldsymbol{u}$ is the basal ice velocity, $n=3$ is a power-law exponent, $C$ is the basal shear stress factor, and $\kappa$ is an empirical coefficient. The effective pressure $N$ is computed with a parameterization from Leguy et al. (2014):

$$N\left(p\right) = \rho_i g H \left(1 - \frac{H_f}{H}\right)^p, \tag{2}$$

where $\rho_i$ is ice density, $g$ is gravitational acceleration, $H$ is the ice thickness, $H_f = \max(0, -\frac{\rho_o}{\rho_i}b)$ is the flotation thickness, $\rho_o$ is ocean water density, and $b$ is the bed elevation, defined as negative below sea level. The parameter $p$ represents the hydrological connectivity of the subglacial drainage system to the ocean and varies between zero (no connectivity) and one (strong connectivity). With $p=0$, the effective pressure is equal to the ice overburden pressure, $N = \rho_i g H$, and the basal shear stress decreases abruptly to zero at the grounding line. With $p=1$, the effective pressure $N = \rho_i g(H - H_f)$, implying full water-pressure support from the ocean wherever $b < 0$. For any $p > 0$, we have $N = 0$ at the grounding line (where $H = H_f$), and therefore the basal stress is continuous across the grounding line. In this case, the transition between Weertman and Coulomb behaviors is smooth and is consistent with basal cavitation during flow over a hard, bumpy bed (Schoof, 2005; Gagliardini et al., 2007).

Equation (1) has two asymptotic behaviors. In most of the ice sheet, where $\kappa|\boldsymbol{u}| << N^n$, Eq. (1) reduces to a Weertman power law (Weertman, 1972):

$$\boldsymbol{\tau_b} = C|\boldsymbol{u}|^{\frac{1}{n}-1}\boldsymbol{u}. \tag{3}$$

If $p = 0$, then Eq. (1) reduces to Eq. (3) everywhere except where the ice is very thin. For $p > 0$, we have $N$ decreasing toward the grounding line, where the ice is thin and fast-flowing. Near the grounding line, $\kappa|\boldsymbol{u}| >> N^n$, and Eq. (1) asymptotes to a Coulomb friction law (Schoof, 2005):

$$\boldsymbol{\tau_b} = \frac{C}{\kappa^{1/n}}N\frac{\boldsymbol{u}}{|\boldsymbol{u}|}. \tag{4}$$

Below, we will apply Eqs. (1) and (2) with different values of $p$ and compare the results to those obtained with a power law, Eq. (3).

## 2.2 Grounding-line parameterization for basal friction

Near the grounding line, GLPs typically compute the basal friction as an area-weighted average of the (possibly large) friction beneath grounded ice and zero friction beneath floating ice. CISM's GLP is similar to the PA_GB1 scheme of Gladstone et al.

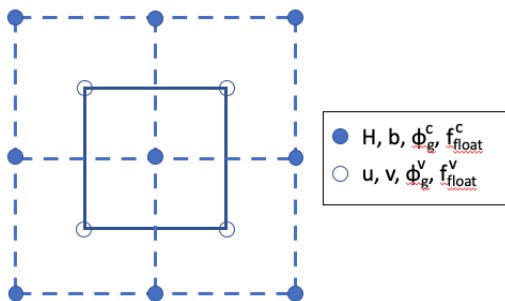

**Figure 1.** Schematic of the CISM grid with ice thickness $H$ and bed topography $b$ located at cell centers (filled circles), and velocity components $(u, v)$ at cell vertices (unfilled circles). The grounded ice fraction $\phi_g$ and flotation function $f_{\text{float}}$ can be computed at either centers or vertices as denoted by superscripts $c$ and $v$, respectively.

(2010), extended to two dimensions. Ice is deemed to be floating if it satisfies a flotation condition:

$$\rho_i H < (-\rho_o b). \tag{5}$$

Based on this condition, we define a flotation function that depends linearly on $H$ and $b$:

$$f_{\text{float}} = -b - \frac{\rho_i}{\rho_o} H, \tag{6}$$

which gives $f_{\text{float}} = 0$ at the grounding line, $f_{\text{float}} < 0$ for grounded ice, and $f_{\text{float}} > 0$ for floating ice. When positive, $f_{\text{float}}$ is equal to the depth of the ocean cavity between the ice-shelf base and the seafloor.

Given $f_{\text{float}}^c$ at cell centers, CISM uses bilinear interpolation to compute a grounded ice fraction $\phi_g^v$ at vertices (where the superscripts $c$ and $v$ denote quantities at cell centers and vertices, respectively; see Fig. 1). The value of $\phi_g^v$ lies in the range
$[0, 1]$ and is set to the grounded fraction in the rectangular bounding box (i.e., staggered grid cell) that surrounds the vertex. The corners of this box are the four neighboring cell centers, each with a value of $f_{\text{float}}^c$ that determines whether the ice at that point is floating. The number of floating cells $n_{\text{float}}$ surrounding a vertex can be 0, 1, 2, 3 or 4. For $n_{\text{float}} = 0$, the ice at the vertex is fully grounded with $\phi_g^v = 1$. Similarly, $n_{\text{float}} = 4$ implies fully floating ice with $\phi_g^v = 0$. In the intermediate cases $n_{\text{float}} = 1, 2$ or 3, the grounding line passes through the bounding box, giving $0 < \phi_g^v < 1$. There are three configurations to consider:

1. One cell is grounded and three are floating (or vice versa). In this case the grounding line passes through two adjacent edges of the box.

2. Two adjacent cells (which share an edge) are grounded, and the other two cells are floating. In this case the grounding line passes through two opposite edges of the box.

3. Two cells at opposite corners of the box are grounded, and the other two are floating. In this case the grounding line
passes through all four edges of the box, with one segment passing through two adjacent edges and another segment passing through the other two adjacent edges.

We compute $\phi_g^v$ as the fraction of the bounding box for which $f_{\text{float}} < 0$. Within this box, $f_{\text{float}}$ can be written as a bilinear function of $x$ and $y$:

$$f_{\text{float}}(x,y) = a + bx + cy + dxy, \tag{7}$$

where $x$ and $y$ are scaled to vary between 0 and 1. With the southwest corner at $(0,0)$ and the northeast corner at $(1,1)$, the coefficients are

$$
\begin{aligned}
a &= f^{SW}, \\
b &= f^{SE} - f^{SW}, \\
c &= f^{NW} - f^{SW}, \\
d &= f^{NE} + f^{SW} - f^{NW} - f^{SE},
\end{aligned}
\tag{8}
$$

where we have dropped the subscript "float" and replaced the superscript $c$ with compass directions.

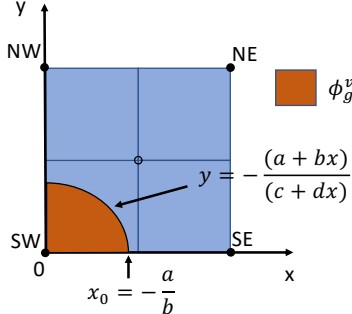

**Figure 2.** Example computation of the grounded area fraction in the staggered grid cell defined by four cell centers (filled circles) adjacent to a cell vertex (hollow circle). Based on a flotation condition, Eq. 5, the lower left cell center (SW) is grounded, and the other three cell centers (SE, NE, NW) are floating. The curved black line is the grounding line, where $f_{\text{float}}(x,y) = a + bx + cy + dxy = 0$, with coefficients given by Eq. (8). The grounded area fraction, $\phi_g^v$, is found by integrating the area between the grounding line and the SW cell center, which lies at the origin (0,0) in a local coordinate system.

As an example, suppose the southwest cell center is grounded and the other three cell centers are floating (Fig. 2). The grounding line passes through the west and south edges of the bounding box. Along the south edge, $y = 0$ and therefore $f = a + bx$. Setting $f = 0$, we find that the grounding line intersects the south edge at $x_0 = -a/b$. The equation for the grounding line is $a + bx + cy + dxy = 0$, from which it follows that $y = -(a+bx)/(c+dx)$ along the grounding line. Then $\phi_g^v$ is equal to the fractional area $\phi$ of the grounded southwest region of the cell, which can be obtained analytically (e.g., using an online integral calculator):

$$\phi_g^v = \int_0^{x_0} y(x)\,dx = \frac{(bc-ad)\ln\left|1-\frac{ad}{bc}\right| + ad}{d^2}. \tag{9}$$

When $ad = bc$, the integral reduces to

$$\phi_g^v = \frac{a^2}{2bc}.$$ (10)

The integrals for other configurations of floating and grounded cells can also be found analytically.

A variation of this method is first to interpolate $f_{\text{float}}^c$ to the four corners of each quadrant, and then to compute $\phi_g$ for each quadrant. A sum over the four quadrants in Fig. 2 yields $\phi_g^v$ for the staggered grid cell. A similar sum over the four quadrants surrounding each cell center yields $\phi_g^c$, the grounded fraction in an unstaggered grid cell (which is used in the basal melt GLP discussed in Section 2.3). When applying basal friction and basal melt GLPs together, CISM uses the quadrant method, which ensures that the global sums of grounded and floating ice area are consistent on the staggered and unstaggered grids.

The grounded fraction $\phi_g^v$ enters the ice dynamics via the basal friction coefficient $\beta$, defined as the ratio of basal shear stress to basal velocity:

$$\tau_{\mathbf{b}} = \beta \mathbf{u}$$ (11)

For example, we would have $\beta = C|\boldsymbol{u}|^{\frac{1}{n}-1}$ for a Weertman law, Eq. (3), and $\beta = CN/(\kappa^{1/n}|\boldsymbol{u}|)$ for a Coulomb law, Eq. (4). After $\beta$ is found at each vertex using the chosen sliding law, it is multiplied by $\phi_g^v$, and the result is used in the basal friction terms of the velocity solver. Since $\phi_g^v$ varies smoothly and linearly with $H$ and $b$ in neighboring cells, the basal friction varies smoothly as the grounding line migrates, improving the behavior of the numerical solution.

## 2.3 Grounding-line parameterization for basal melting

As discussed by Seroussi and Morlighem (2018), marine ice-sheet dynamics can be sensitive to the way basal melt is applied near the grounding line. If melting is applied to a partly grounded grid cell, then part of the melt rate is effectively used to melt grounded ice, which can lead to excessive thinning and grounding-line retreat. According to theory (Schoof, 2007a), an unbuttressed, two-dimensional ice shelf in steady state can be melted completely from below without any change in grounding-line location. Thus, a melt scheme should not trigger grounding-line retreat when applied to an unbuttressed, steady-state shelf. For three-dimensional shelves with buttressing, the dynamics are more complex. On a retrograde bed (i.e., a bed that slopes upward in the direction of ice flow), grounding lines are unconditionally unstable, assuming no flow variation in the transverse direction. Buttressing, however, can stabilize the grounding line on retrograde beds (Gudmundsson, 2013). Thus, it is not obvious which treatment is best for a buttressed ice shelf: a scheme that applies no melt in cells containing the grounding line, or a scheme that allows some melt in partly grounded cells.

In CISM, cells that are partly grounded and partly floating have $0 < \phi_g^c < 1$, where $\phi_g^c$ is the grounded fraction at cell centers (instead of vertices as in Sect. 2.2). We have implemented, and will test below, three ways of computing basal melt in such cells. In each case, we start with a baseline melt rate $m$, which might be a function of horizontal location or depth and is applied to cells that are fully floating. In cells that are partly grounded, we compute an adjusted value $m_{\text{GLP}}$. The three options are as follows:

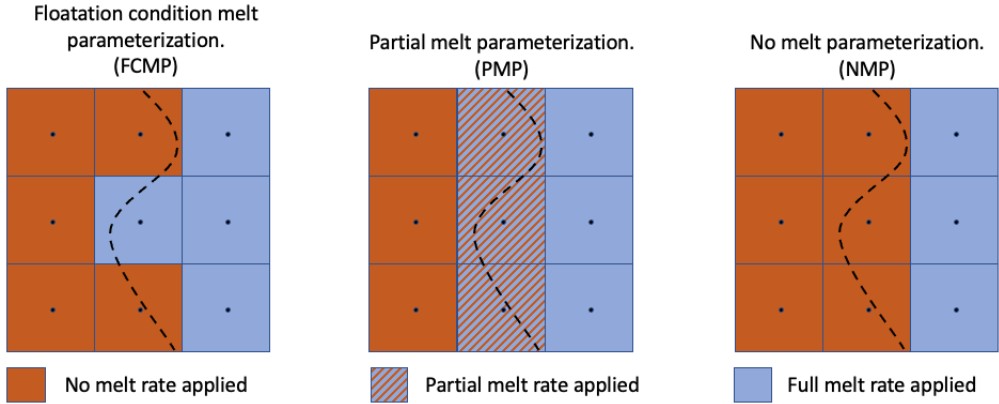

**Figure 3.** The different melt parameterizations applied in cells containing the grounding line. The grounding line (black dashed line) can be located in cells where ice is considered fully grounded and where no melt rate is applied (brown), in cells where ice is fully floating and where the full melt rate is applied (blue), and in cells that are partially floating/grounded, where a partial melt rate can be applied (blue with brown stripes). We consider three melt rate parameterizations. Cells containing the grounding line experience one of the following: no melt or full melt depending on a flotation criterion (FCMP, left), partial melt proportional to the fraction of the cell that is floating (PMP, middle), or no melt (NMP, right). Black circles show cell centers, where the flotation criterion is applied for the FCMP.

1. At each grid cell center, determine whether the cell satisfies the flotation condition, Eq. (5). If the cell is floating, set $m_{\mathrm{GLP}} = m$. Otherwise, set $m_{\mathrm{GLP}} = 0$. We refer to this option as the Flotation Condition Melt Parameterization (FCMP). It is illustrated in the left panel of Fig. 3, where the grounding line passes through the central column of three cells. One cell (shaded blue) satisfies Eq. (5) and receives full melt, while the other two cells (shaded brown) receive no melt.

2. In partly floating grid cells, compute the floating fraction $\phi_f^c = 1 - \phi_g^c$, and set $m_{\mathrm{GLP}} = \phi_f^c m$. This is the Partial Melt Parameterization (PMP), shown in the central panel of Fig. 3 by blue and brown striping in cells where the melt is proportional to $\phi_f^c$.

3. In all partly grounded cells, set $m_{\mathrm{GLP}} = 0$. Following Seroussi and Morlighem (2018), we call this the No Melt Parameterization (NMP). It is illustrated in the right panel of Fig. 3, where the partly grounded cells are shaded brown and have no melt.

Of these three melt GLPs, only PMP ensures that the melt rate varies continuously with subgrid changes in grounding-line location, as determined by the condition $f_{\mathrm{float}} = 0$ using Eqs. (6)–(8). PMP is similar to the Sub-Element Melt 1 (SEM1) method of Seroussi and Morlighem (2018), but there is no equivalent in their study to FCMP. We also considered a "Full Melt Parameterization" in which the full basal melt rate is applied in partly grounded cells. In preliminary tests, we found that this scheme drives unrealistic grounding-line retreat in CISM, as in Seroussi and Morlighem (2018), and we will not consider it further.

## 3   CISM testing using the MISMIP3d experiments

The results described in this section are based on the MISMIP3d experiments (Pattyn et al., 2013), which are designed to study the effect of a spatially variable perturbation in basal shear stress in marine ice-sheet models. Leguy et al. (2014) used a one-dimensional model to explore the effect of different basal friction laws on grounding-line migration. They found that a resolution of ~1 km (and, in some circumstances, coarser) is sufficient to accurately simulate grounding-line motion if the ice sheet is hydrologically well connected to the ocean, or if a GLP is used for basal shear stress. In contrast, ice sheet models without hydrological connectivity or a GLP (including models that participated in MISMIP3d, as shown in Fig. 5 of Pattyn et al. 2013) require very high resolution ($\sim 500$ m or finer) to accurately capture grounding-line dynamics. This 1D model was practical for running many experiments at low computational cost, but there was no guarantee that the results would generalize to three-dimensional models. We now use CISM for this purpose.

### 3.1   Experimental setup

Here, we briefly recall the experimental setup for MISMIP3d. A full description can be found in Pattyn et al. (2013). The goal of the experiment is to analyze the transient behavior of the grounding line when subject to a spatially varying perturbation in basal stress parameters. The domain is rectangular, 800 km long in $x$, 100 km wide in $y$, with $x = 0$ at the left-hand side of the box and $y = 0$ at the center. The bed is linear with no lateral variation, sloping downward toward the ocean:

$$b(x,y) = \left( -100 - \frac{x}{1\text{ km}} \right)\text{m}. \tag{12}$$

The ice flows freely at side boundaries.

**Table 1.** Parameter values for the MISMIP3d experiments.

| Parameters | Value | Units | Definition |
|---|---|---|---|
| $a$ | 0.5 | m a$^{-1}$ | Accumulation rate |
| $\rho_{\text{i}}$ | 900 | kg m$^{-3}$ | Ice density |
| $\rho_{\text{w}}$ | 1000 | kg m$^{-3}$ | Seawater density |
| $A$ | $3.1536 \times 10^{-18}$ | Pa$^{-3}$ a$^{-1}$ | Ice softness |
| $C$ | 31651.76 | Pa m$^{-1}$ a$^{1/3}$ | Shear stress factor |
| $g$ | 9.8 | m s$^{-2}$ | Gravitational acceleration |
| $n$ | 3 | – | Glen's flaw-low exponent |
| $\kappa$ | $2.5 \times 10^{15}$ | Pa$^3$ a m$^{-1}$ | bed characteristic empirical coefficient |
| $W$ | 50 000 | m | Half-width of domain |
| | 31 536 000 | s a$^{-1}$ | Seconds per year |

Given the parameters in Table 1, the model is initialized using a semi-analytical ice-sheet profile from Schoof (2007a) with a grounding line located at $x \approx 250$ km. The experiment then consists of three steps, which we denote as in Pattyn et al. (2013).

In the first step, labeled "Stnd", the model is initialized with a uniform 500-m thick slab of ice and is spun up over 20,000 years with a uniform basal shear stress factor. At the end of the spin-up, the ice sheet has reached a steady state in which the change in grounding line location is less than $10^{-3}$ m a$^{-1}$ and the change in ice thickness at the grounding line is less than $10^{-4}$ m a$^{-1}$. Next, in a step labeled "P75S", the shear stress factor is modified to vary spatially and the model is run forward in time for 100 years. The shear stress factor perturbation is defined by a Gaussian bump:

$$C^{\star} = C \left( 1 - a \exp \left[ -\frac{(x - x_b)^2}{2x_c^2} - \frac{(y - y_b)^2}{2y_c^2} \right] \right), \tag{13}$$

where $C^{\star}$ is the perturbed shear stress factor, $a = 0.75$ is the perturbation amplitude, $x_b$ is the initial grounding-line position at $y = 0$ km, $y_b = 0$ km, $x_c = 150$ km, and $y_c = 10$ km. This perturbation increases the slipperiness near the grounding line at the center of the domain.

After 100 years, the shear stress factor is reset to its original value, and the model is again run to steady state. This step, labeled "P75R", tests whether the model dynamics is reversible, as predicted by theory (Schoof, 2007a). In other words, by the end of P75R, the grounding-line should return to its position at the end of Stnd.

This setup allows flowline models using the SSA with $p = 0$ (as defined in Eq. (2)) to be compared with the semi-analytical solution of Schoof (2007a) in the unperturbed initial and final steady-state configurations.

## 3.2 Steady-state results

Using the MISMIP3d setup, we investigate the effects of CISM's basal sliding law (see Sect. 2.1) and basal friction GLP (see Sect. 2.2) in experiments using the SSA, DIVA, and BP velocity solvers. We show results for six model configurations: with and without a GLP, and for $p = 0$, 0.5 and 1. For SSA and DIVA, we test five grid resolutions (8, 4, 2, 1, and 0.5 km), and for BP we test four resolutions (omitting 0.5 km because of the high computational cost).

Figure 4 shows the steady-state grounding line position as a function of resolution at the end of the Stnd experiments for SSA, DIVA, and BP, with $p = 0$, 0.5, and 1, both with and without a GLP. When the grounding-line position no longer changes significantly as resolution is increased, we consider the solution to have converged. The threshold for "significantly" depends on the application, but without stating a specific threshold, we can see that for $p \leq 0.5$, a GLP greatly improves convergence. Using DIVA with a GLP, for example, the 1-km solution differs from the 0.5-km solution by about 6 km with $p = 0$ and by 9 km with $p = 0.5$. Without a GLP, the respective differences are 37 km and 28 km. With $p = 1$, however, a GLP does not clearly improve convergence. These results are in agreement with Leguy et al. (2014).

The experiments using the SSA with $p = 0$ (upper left of Fig. 4) can be compared directly with the semi-analytic solution from Schoof (2007a). This solution approximates the SSA and is not an exact diagnosis of the grounding line, as it neglects longitudinal stresses except in the boundary layer very near the grounding line. While it provides a good comparison benchmark, small differences should be expected when comparing with converged numerical solutions. In the MISMIP3d experiments, the semi-analytic solution puts the grounding line at $x \approx 611$ km. With a GLP, the grounding line lies at 600.0, 604.0 and 605.5 km with resolutions of 2 km, 1 km, and 0.5 km, respectively. Without a GLP, the grounding line does not advance far enough, falling $\sim 100$ km short of the semi-analytic solution even at 0.5 km resolution.

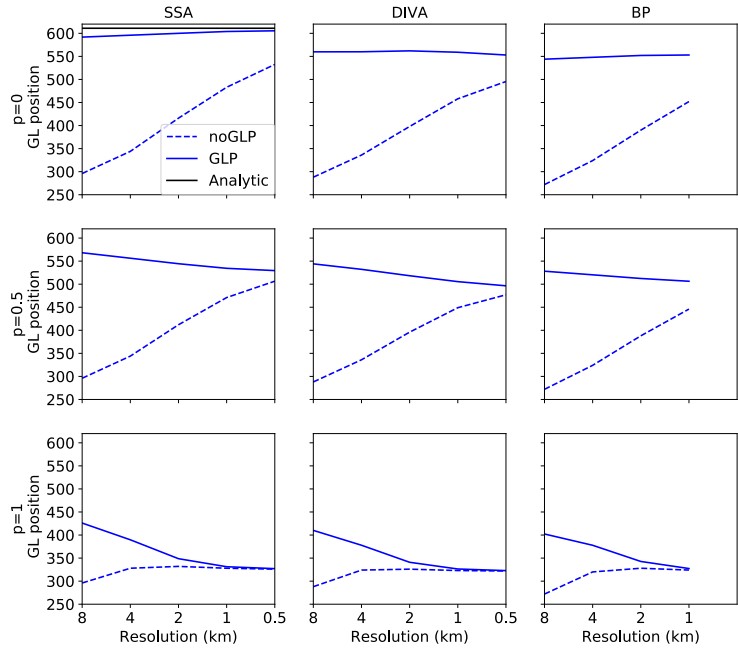

**Figure 4.** Location of the central grounding line (km) as a function of grid resolution (km) at the end of the MISMIP3d Stnd experiments for SSA (left), DIVA (middle), BP (right), and using $p = 0$ (top), $p = 0.5$ (middle), $p = 1$ (bottom), without a GLP (dashed lines) and with a GLP (solid lines). The semi-analytic solution from Schoof (2007a) is shown in black for SSA and $p = 0$.

Once converged, the difference in grounding-line position between SSA and BP depends strongly on $p$. For $p = 0$, the difference is ~50 km, since vertical shear (captured by BP but not SSA) is significant when the basal stress near the grounding line is large. When $p = 1$ and the bed is slippery, the SSA and BP grounding lines are only 5 km apart. Unlike the SSA, the converged DIVA solutions are close to the converged BP solutions for all values of $p$. When $p \leq 0.5$, the difference in grounding line position between DIVA and BP is within 10 km. The differences are within 2 km with $p = 1$, where the increased bed

lubrication reduces the importance of resolving vertical shear. These results suggest a relatively large gain in accuracy (at least for smaller values of $p$) when moving from SSA to DIVA, and a smaller gain when moving from DIVA to BP. Since DIVA is depth-integrated, it is similar in cost to SSA. For these simulations, DIVA is 10 to 40 times faster than BP, depending on resolution.

     For DIVA, the difference in grounding-line position between 1-km and 0.5-km resolution is 9 km or less, depending on $p$.

This difference is small compared to the ~200-km difference between using DIVA with $p = 0$ and $p = 1$. The $p$ value also has more impact on grounding-line position than does resolution for SSA and BP. These results suggest that for problems similar to MISMIP3d, the uncertainties associated with basal friction near the grounding line are likely to be much greater than those associated with the grid resolution (if using a GLP) and stress approximation (especially if choosing between BP and DIVA). Thus, work is needed to better understand and simulate the hydrological state and effective pressure of marine ice sheets,

especially near the grounding line.

### 3.3 Transient results

Next, we discuss results from the transient experiments, P75S and P75R. At the end of P75R, the grounding line should retreat to its original unperturbed position from the end of the Stnd experiment. We investigate this reversibility as a function of resolution, values of $p$, and Stokes approximations. If the grounding line returns to within 4 km of the unperturbed grounding line position, the experiment is considered reversible. The results for reversibility are shown in Fig. 5.

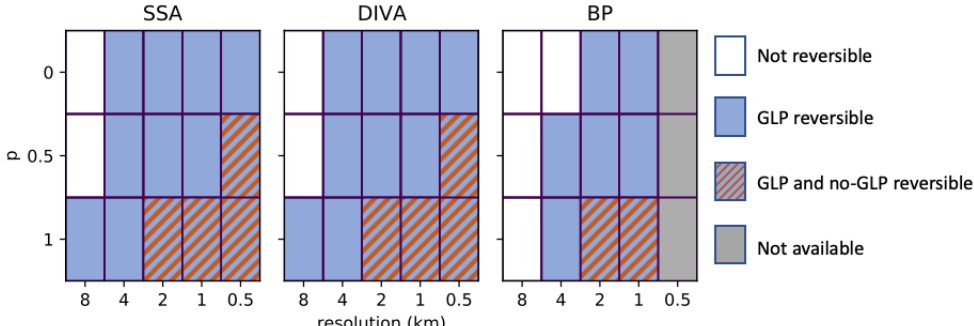

**Figure 5.** Results of grounding-line reversibility tests as a function of resolution in the MISMIP3d P75R experiments for SSA (left), DIVA (middle), BP (right), and using $p = 0$ (top), $p = 0.5$ (middle), $p = 1$ (bottom). Each experiment is either not reversible (white), only reversible with a GLP (blue), reversible with and without a GLP (blue-brown striping), or not available (grey) because the run was not performed at that resolution. An experiment is considered reversible if the grounding line at the end of P75R retreats to within 4 km of its unperturbed position at the end of experiment Stnd.

As for Fig. 4, these results show the importance of using a GLP. Without a GLP, the grounding line is reversible only when either the resolution is very high (0.5 km) or the bed is slippery near the grounding line ($p = 1$). With a GLP, reversibility is achieved at 4-km resolution or finer for nearly all stress approximations and values of $p$. Even for the 8-km experiments that do not meet our threshold for reversibility, the grounding line usually returns within one grid cell (i.e., 8 km) of its original location.

Since the 4-km threshold for reversibility is somewhat arbitrary, we note that the results are not very sensitive to this threshold. With a 2-km threshold, the SSA and DIVA tests with $p = 0$ are labeled as irreversible at 4-km resolution. When the threshold is reduced to 1 km, the BP test with $p = 0$ at 2-km resolution becomes irreversible. Otherwise, Fig. 5 is unchanged.

The first column of Fig. 6 illustrates the three-part sequence of MISMIP3d experiments, showing the grounding line location at the end of Stnd, P75S, and P75R for $p = 0, 0.5$, and 1, using DIVA with a GLP at 1-km resolution. Experiment P75S shows that, as $p$ increases, the central region of grounding-line advance becomes narrower, with greater retreat along the channel boundaries. This behavior is seen at all resolutions and for SSA and BP (not shown) as well as DIVA. Columns 2–4 of Fig. 6 show, for each approximation and resolution, the maximum grounding line displacement in P75S relative to the end of Stnd. Consistent with the reversibility results, these transient solutions show better convergence with resolution when using a GLP, especially for $p = 0$ and 0.5. With a GLP and $p \leq 0.5$, the maximum advance is several kilometers less with SSA than with

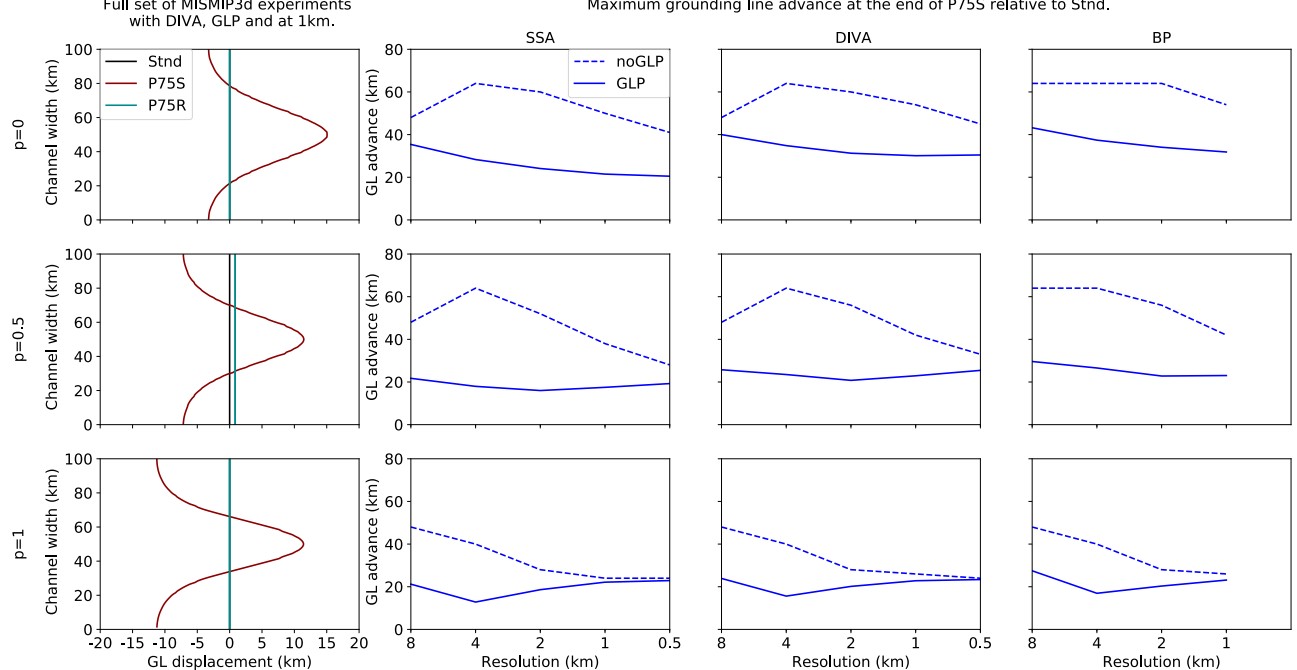

**Figure 6.** Grounding-line profiles for the MISMIP3d experiments. The three rows show results with $p = 0$ (top), $p = 0.5$ (middle), and $p = 1$ (bottom). The first column shows the grounding line displacement (km) relative to the end of experiment Stnd for the full set (Stnd (black), P75S (red), P75R (cyan)) of MISMIP3D experiments using DIVA with a GLP, at a resolution of 1 km. The Stnd and P75R positions are visually indistinguishable for $p = 0$ and $p = 1$. Columns 2–4 show the most advanced grounding line location at the end of experiment P75S, relative to the end of Stnd, as a function of grid resolution. Columns 2–4 are arranged as in Fig. 4.

DIVA and BP. Compared to the initial steady-state solution, the maximum transient displacement is a relatively weak function of Stokes approximation and $p$. This weak dependence, however, might depend on the nature of the perturbation.

At coarser resolution (not shown), the grounding line retreats a few km less at the channel boundaries than shown here at 1-km resolution. Without a GLP and with $p \leq 0.5$, the grounding line does not retreat anywhere during P75S, but advances across the entire channel. This is why the maximum advance is greater without a GLP than with a GLP. This absence of retreat along the boundaries is also seen with some models participating in MISMIP3d (e.g., TAL3 and SCO0, as shown in Fig. 5 of Pattyn et al. 2013), using a resolution of 1 km or coarser and no GLP. For models using either high resolution at the grounding line (e.g., DMA6, RHI1, and SCO6) or a GLP at 1-km resolution (TAL1), the results in Pattyn et al. (2013) are similar to our results with $p = 0$ and a GLP.

Finally, we examine grounding-line evolution on decadal time scales in MISMIP3d. Figure 7 shows the transient grounding-line displacement for experiment P75S and the first 100 years of P75R relative to the position at the end of Stnd. Results are shown for the center ($y = 0$) and upper boundary ($y = 50$ km) of the domain using DIVA with a GLP for $p = 0$, 0.5, and 1, at resolutions of 1, 2, and 4 km. For P75S, the grounding line motion depends only weakly on resolution at a given value of

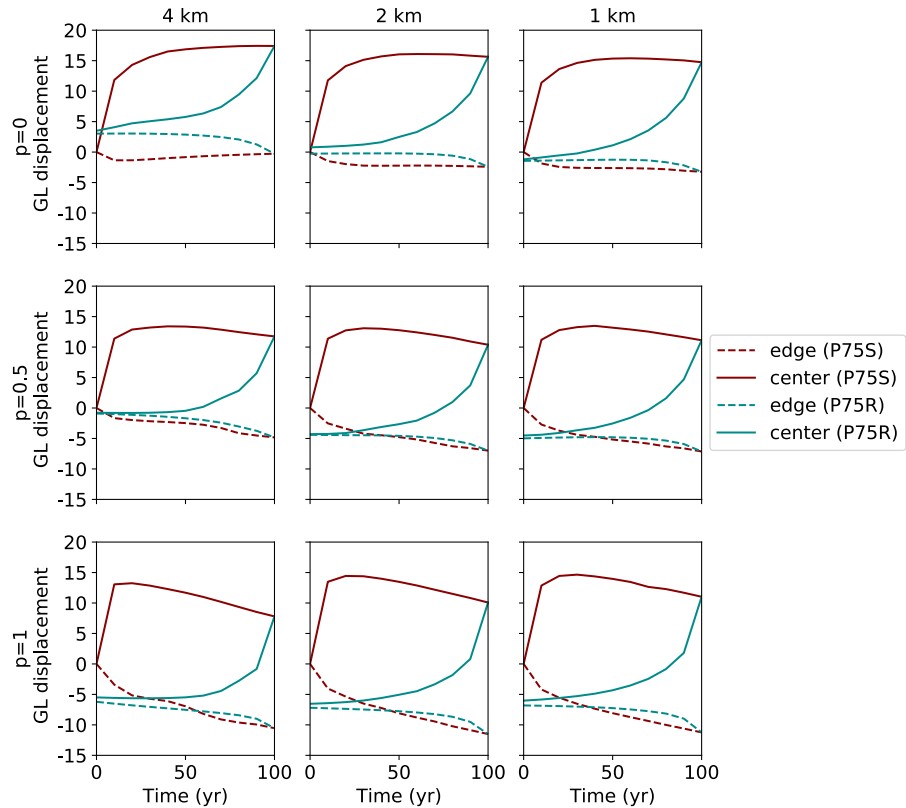

**Figure 7.** Grounding line displacement relative to the end of MISMIP3d experiment Stnd for the first 100 years of experiments P75S (red) and P75R (cyan), at the center (solid line) and edge (dashed line) of the domain, and at resolutions of 4 km (left), 2 km (center), 1 km (right), for $p = 0$ (top), $p = 0.5$ (middle), $p = 1$ (bottom). The DIVA stress approximation with a GLP was used for these experiments.

$p$. The maximum advance at the center of the domain is about 15 km for all $p$-values, but the retreat at the edge is greater for larger $p$. For all $p$ values and resolutions, the central advance is greatest during the first 10–20 years, after which it levels off ($p = 0$) or decreases slowly ($p > 0$). The central grounding line is still retreating after 100 years. At the edge, there is rapid retreat for 10–20 years, followed by leveling off ($p = 0$) or slower retreat ($p > 0$).

During P75R, for all $p$ values and resolutions, the grounding line retreats at the center and advances at the edges, most rapidly during the first 10–30 years. The grounding line retreats to within a few kilometers of its original position after 100 years, but takes longer to reach equilibrium. For $p = 0$, finer resolution yields a retreat that is more nearly complete, but this is not true for $p > 0$. It is not clear from these experiments under what conditions the retreat to full reversibility would take place on the same time scale as the duration of the perturbation.

In summary, we have shown that the conclusions of Leguy et al. (2014) carry over to a three-dimensional model using the MISMIP3d framework. For all approximations, a GLP improves accuracy and reversibility at a given resolution, especially for $p = 0$ and $p = 0.5$. The depth-integrated DIVA solver gives results similar to BP at a small fraction of the computational cost.

## 4 Ice shelf melt experiments

Next, we analyze three numerical schemes for sub-shelf melting near the grounding line: FCMP, PMP, and NMP, as described in Sect. 2.3. We apply these schemes first to an unbuttressed shelf, as in MISMIP3d, and then to a buttressed shelf, as in MISMIP+ (Asay-Davis et al., 2016).

Theory predicts that at steady state, melting of an unbuttressed SSA ice shelf should not affect the upstream grounded ice, and the grounding line should remain stationary (Schoof, 2007a). This requirement is most easily satisfied with NMP, since both PMP and FCMP apply some melt and thinning in partly grounded cells, potentially driving grounding-line retreat. To test the three schemes, we initialize the ice shelf for each $p$-value and resolution, for both SSA and DIVA, as in the Stnd experiment described in Sect. 3.1. Starting from steady state, we apply a uniform sub-shelf melt rate of 30 m a$^{-1}$ for 1000 a and measure

the grounding-line retreat, if any. Results are described here but are not shown, since in most cases there is little change in the grounded ice thickness or grounding-line location over 1000 a.

     With NMP, as expected, grounding-line retreat is very small: tens of meters when $p = 0$ and up to a few hundred meters otherwise. With FCMP and PMP, the retreat is slightly larger, but is always less than one grid cell, provided the resolution is 4 km or finer. This is true for both SSA and DIVA, and for all $p$ values. At 8 km, using FCMP and PMP, there are three cases

(p = 0.5 with SSA and DIVA, and p = 1 with SSA) in which the grounding line retreats by up to four grid cells in 1000 a. This result suggests that 8-km resolution is too coarse to apply partial melt at the grounding line. At 4 km or finer, however, the minimal response suggests that FCMP and PMP can safely be applied to unbuttressed ice shelves without driving spurious grounding-line retreat.

     In the rest of this section, we consider a set of ice shelf melt experiments using the MISMIP+ experimental setup, which

provides a more realistic framework for studying marine ice sheets by adding lateral buttressing and sub-shelf melting. We adopt this setup as described by Asay-Davis et al. (2016), but with a depth-dependent melt rate from Seroussi and Morlighem (2018). We analyze the model response to the three melt schemes FCMP, PMP, and NMP. As we show below, our results differ in important respects from those of Seroussi and Morlighem (2018). We carry out several sets of experiments with melt-induced ice retreat:

– experiments with a moderately large melt rate, following Seroussi and Morlighem (2018);

     – experiments with a high melt rate, five times larger than the moderate melt rate;

     – experiments with a large location-dependent melt rate to imitate calving;

     – experiments with a moderate melt rate as in the first set of experiments, but for an ice shelf flowing about six times slower.

For these melt experiments, as for MISMIP3d, using a GLP is always beneficial, and results from DIVA and BP are very similar. In this framework, in contrast to MISMIP3d, we find that SSA results are also similar to DIVA. This similarity can be explained by the lower shear stress factor $C$ and greater ice softness $A$ in MISMIP+, leading to a faster, less viscous flow

and reduced vertical shear. We will focus on experiments using DIVA with a GLP, applying different resolutions, $p$ values, and melt parameterizations.

## 4.1 Experimental setup

The goal of MISMIP+ (Asay-Davis et al., 2016) is to analyze the transient behavior of an ice stream and attached shelf when subject to calving events or changes in sub-shelf melting. Cornford et al. (2020) analyzed MISMIP+ results from many ice sheet models, including CISM. Here, we use the MISMIP+ framework to study sub-shelf melting near the grounding line.

The domain is a rectangle, 80 km wide in $y$, and 640 km long in $x$. The origin is at the left-hand edge in $x$ and at the center of the domain in $y$. The bedrock topography $b$, similar to that of Gudmundsson (2013), is given by

$$b(x,y) = \max\left[B_x(x) + B_y(y), z_{\text{deep}}\right],$$  (14)

$$B_x(x) = B_0 + B_2\tilde{x}^2 + B_4\tilde{x}^4 + B_6\tilde{x}^6,$$  (15)

$$\tilde{x} = \frac{x}{\bar{x}},$$  (16)

$$B_y(y) = \frac{d_c}{1 + \exp^{-2(y-L_y/2-w_c)/f_c}} + \frac{d_c}{1 + \exp^{2(y-L_y/2+w_c)/f_c}},$$  (17)

and is illustrated in Fig. 8. Table 2 lists the values and definitions of these and other MISMIP+ parameters that differ from the MISMIP3d parameters listed in Tab. 1. In the $y$ direction, the side walls slope down from the edges toward a central trough. The trough, oriented along the $x$-axis, is described by a nonlinear polynomial, unlike the linear bed in MISMIP3d. The bed is retrograde (sloping upward in the direction of ice flow) between $x = 390$ km and $x = 505$ km, and otherwise slopes downward toward the open ocean.

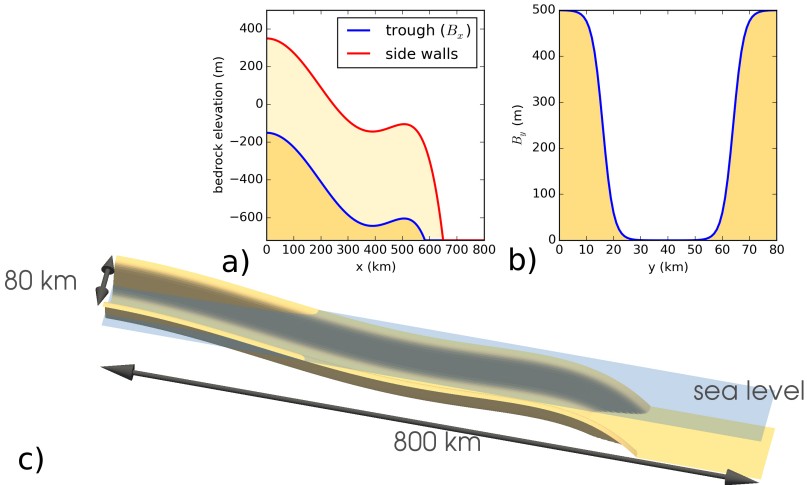

**Figure 8.** The bedrock topography used in the melt rate experiments shown in different views: along the x-direction (a), across the y-direction (b) and the full 3-dimensional view (c). This figure is reproduced from Fig. 1 of Asay-Davis et al. (2016).

**Table 2.** Parameter values for the MISMIP+ experiments.

| Parameters | Value | Units | Definition |
|---|---|---|---|
| $B_0$ | -150.0 | m | Bedrock topography at $x = 0$ |
| $B_2$ | -728.8 | m | Second bedrock topography coefficient |
| $B_4$ | 343.91 | m | Third bedrock topography coefficient |
| $B_6$ | -50.57 | m | Fourth bedrock topography coefficient |
| $\bar{x}$ | 300 | km | Characteristic along-flow length scale of the bedrock |
| $f_c$ | 4.0 | km | Characteristic width of the side walls of the channel |
| $d_c$ | 500 | m | Depth of the trough compared with the side walls of the channel |
| $w_c$ | 24.0 | km | Half-width of the trough |
| $z_{\text{deep}}$ | -720 | m | Maximum depth of the bedrock topography |
| $x_{\text{calve}}$ | 640 | km | The location in $x$ beyond which the ice is removed |
| $a$ | 0.3 | m a$^{-1}$ | Accumulation rate |
| $A$ | $2.0 \times 10^{-17}$ | Pa$^{-3}$ a$^{-1}$ | Ice softness (default value) |
| $\rho_{\text{i}}$ | 918 | kg m$^{-3}$ | Ice density |
| $\rho_{\text{w}}$ | 1028 | kg m$^{-3}$ | Seawater density |
| $C$ | $1 \times 10^4$ | Pa m$^{-1}$ a$^{1/3}$ | Shear stress factor |

MISMIP+ participants can choose among three basal sliding laws: (1) the Schoof (2005) law, Eq. (1); (2) a sliding law given by Tsai et al. (2015), which like Eq. (1) has powerlaw and Coulomb behavior as limits; or (3) a Weertman power law, Eq. (3). The stress-balance approximation is up to the modeler. The value of $A$, the temperature-independent factor in Glen's flow law (Tab. 3), is typically adjusted to obtain a steady-state grounding line in the desired location on the retrograde part of the bed. Our CISM experiments use DIVA with the Schoof sliding law, Eq. (1), unless stated otherwise, and explore the sensitivity of

ice stream behavior using three values of the hydrologic connectivity parameter $p$: 0, 0.5, and 1.

Most MISMIP+ participants (Cornford et al., 2020) submitted results at resolutions of 1 km or finer, but continental-scale models often require coarser resolution to reduce computational cost. For this reason, we examine a range of resolutions from 0.5 km to 8 km. The time step is 0.5 a at all resolutions; using a shorter time step does not significantly change the results.

The ice stream is initialized as a uniform slab with $H = 100$ m and is spun up over 20,000 years to a steady state, with

a surface mass balance of 0.3 m a$^{-1}$. The steady state is determined based on grounding-line location and ice thickness, as described in Sect. 3.1. We perform three spin-ups, one for each value of $p$. During the spin-up, there is no melting at the shelf base, so that the spun-up state is independent of the treatment of basal melt. Starting from the steady-state profile, we then perform six experiments, denoted as Ice0, Ice1r, Ice1ra, Ice1rr, Ice1rax, and Ice1rrx. Ice0 is a 100-year control experiment during which no basal melt is applied. In Ice1r, the ice stream evolves for 100 years with the basal melt rate defined by

Eq. (18) below. Experiment Ice1rr continues Ice1r for another 100 years with the same melt rate. Ice1ra also starts at the end

of experiment Ice1r, but proceeds for 100 years with $m = 0$, to observe relaxation toward the original steady-state profile. Experiments Ice1rrx and Ice1rax are the continuations of Ice1rr and Ice1ra, respectively, for another 800 years.

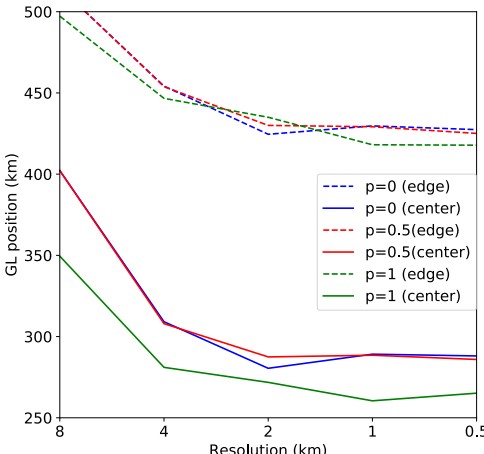

**Figure 9.** Grounding line location (km) as a function of grid resolution (km) at the end of the spin-up, using the MISMIP+ experimental setup. Results are shown for $p = 0$ (blue), $p = 0.5$ (red), and $p = 1$ (green) at the edge (dashed lines) and center (solid lines) of the domain.

Figure 9 shows the grounding line position at the end of the spin-up as a function of resolution and $p$ values, using the DIVA solver and a basal friction GLP with $A = 2 \times 10^{-17}$ Pa$^{-3}$ a$^{-1}$, the default ice softness value for MISMIP+. The position is shown at the center (solid) and edge (dashed) of the domain. The grounding line is farther advanced at the edge, as seen by most MISMIP+ participants (Cornford et al., 2020). For all $p$ and both parts of the domain, the grounding line position is less advanced and more accurate at 4 km than at 8 km. The 4-km solutions lie within 10 km of the 0.5-km solutions at both the center and edge.

Results with $p = 0.5$ are similar to those with $p = 0$. For $p = 1$, however, the grounding line is less advanced by $\sim$10–20 km at the center and (for all resolutions except 2 km) by several km at the edge. The grounding-line position is less sensitive to $p$ in MISMIP+ than in MISMIP3d (Sect. 3.2), in part because some parameters are different in the two experiments. The MISMIP+ parameters make the bed more slippery, so that the transition in basal friction at the grounding line is less abrupt for a given $p$. Also, buttressing provides resistance to flow in MISMIP+ but not MISMIP3d.

MISMIP+ participants (Cornford et al., 2020) were asked to tune their models to begin the perturbation experiments with the grounding line at $450 \pm 10$ km. Similarly, for our perturbation experiments, we tuned $A$ so that the steady-state grounding line lies within 1 km of $x = 455$ km. This value was chosen because it is close to the original grounding-line positions for $p = 0$ and $p = 0.5$ at 0.5 km resolution and therefore required the least retuning. The tuned $A$ values and corresponding grounding-line positions are shown in Tab. 3. Without this tuning, given the nonlinear bed topography, the initial bed slope near the grounding line could be very different at coarse compared to fine resolutions (Fig. 9), which would make it difficult to interpret the response to perturbations. On the other hand, the smaller values of $A$ (i.e., greater viscosity) for $p = 1$ can influence the

**Table 3.** Ice softness $A$ ($10^{-17}\mathrm{Pa}^{-3}\mathrm{a}^{-1}$) and resulting grounding line position $x_g$ (km) for melt experiments at the end of the spin-up. Note that the $A$-values are not truncated.

| | p=0 | | p=0.5 | | p=1 | |
|---|---|---|---|---|---|---|
| Resolution | A | $x_g$ | A | $x_g$ | A | $x_g$ |
| 8 km | 3.5 | 454.64 | 3.5 | 454.63 | 3.1 | 455.27 |
| 4 km | 2.25 | 454.51 | 2.35 | 455.70 | 1.866069 | 454.36 |
| 2 km | 1.95 | 455.38 | 2.0 | 454.99 | 1.9 | 454.42 |
| 1 km | 2.0 | 455.67 | 2.0 | 455.41 | 1.775 | 455.44 |
| 0.5 km | 2.0 | 455.23 | 2.0 | 454.37 | 1.77 | 454.33 |

transient response, and therefore the different responses with $p = 1$ relative to $p \leq 0.5$ must be attributed to differences in both bed lubrication and viscosity.

## 4.2 Moderate basal melt rate experiments

Using the MISMIP+ protocol, we now test the sensitivity of basal melt rates to grid resolution, to determine the resolution needed for model results to converge with different $p$ values and melt parameterizations. We do not use the melt rate profile suggested in Asay-Davis et al. (2016), since it does not allow melt near the grounding line (Dutrieux et al., 2013). Instead, we use a linear, piecewise-continuous, depth-dependent melt rate as in Seroussi and Morlighem (2018), defined as:

$$
m = \begin{cases} 0 \text{ m a}^{-1}, & z_d > -50 \text{ m}, \\ 1/15\,(z_d + 50) \text{ m a}^{-1}, & -500 \text{ m} < z_d < -50 \text{ m}, \\ 30 \text{ m a}^{-1}, & z_d < -500 \text{ m}, \end{cases} \tag{18}
$$

where $z_d$ is the ice shelf basal elevation, and $m$ is defined as positive for melting. This melt profile is characteristic of warm-water shelves, as found in the Amundsen Sea region (Dutrieux et al., 2013; Rignot et al., 2013).

We consider the perturbation experiments Ice1r, Ice1rr and Ice1ra, comparing results for the three basal melt parameterizations described in Sect. 2.3: FCMP, PMP, and NMP. For each experiment we show the change in ice mass above flotation (IMAF), which is often used in continental-scale studies to diagnose sea level change. Changes in grounded ice area are not shown, but are qualitatively similar to changes in IMAF.

Figure 10 shows the evolution in IMAF relative to the spun-up steady state. Results are shown for $p = 0$, 0.5, and 1, for each melt parameterization, and for resolutions 0.5 to 8 km. Experiments Ice1rr and Ice1ra diverge after year 100, with IMAF decreasing further during Ice1rr but stabilizing and starting to recover during Ice1ra. During Ice1r and Ice1rr, for a given $p$-value and melt parameterization, IMAF decreases faster with finer resolution. Results using FCMP and PMP, the two schemes allowing some melt in cells at the grounding line, are very similar to each other, for all $p$ and all resolutions.

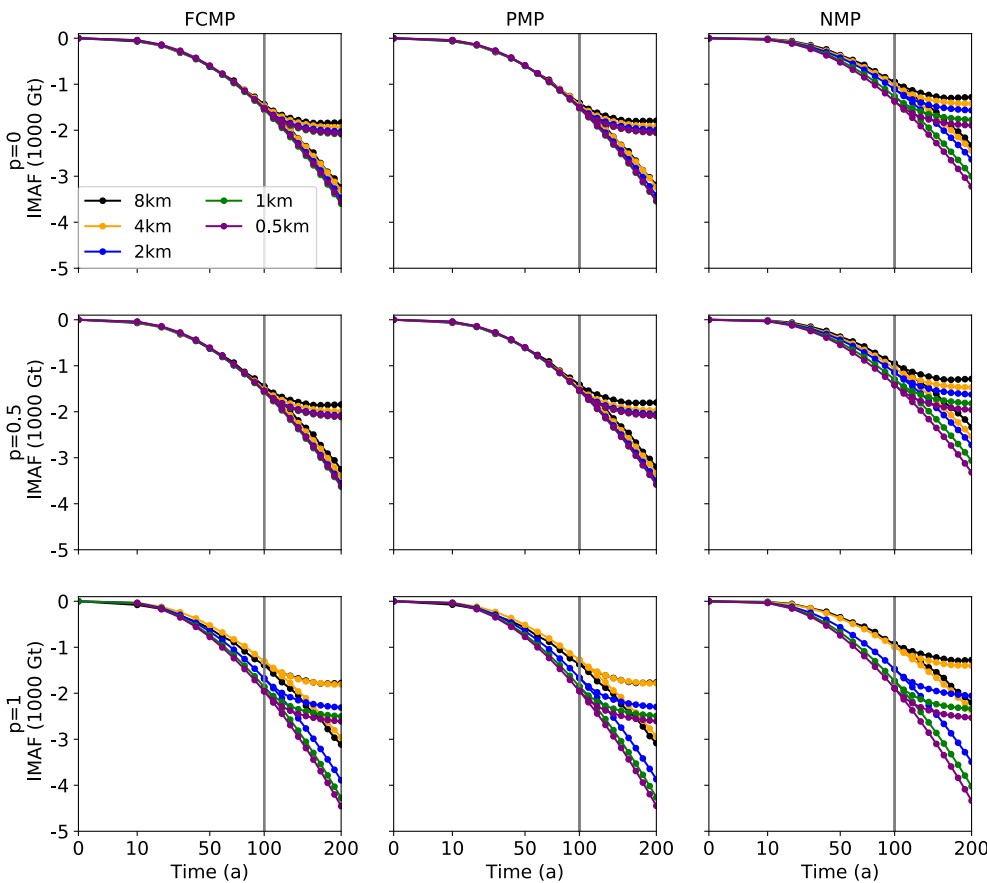

**Figure 10.** Evolution of IMAF (in units of 1000 Gt) relative to steady state as a function of time for the moderate-melt ice shelf experiments Ice1r, Ice1rr, and Ice1ra. Results are shown as a function of $p$ in rows, as a function of melt parameterizations in columns, and at resolutions of 8 km (black), 4 km (orange), 2 km (blue), 1 km (green), and 0.5 km (purple). The vertical line at $t = 100$ a indicates the end of experiment Ice1r and the beginning of experiments Ice1ra (during which melting is turned off) and Ice1rr (during which the melting is unchanged).

Table 4 shows IMAF loss after 200 years for Ice1rr as a function of $p$ and melt parameterization. When $p \leq 0.5$ and at resolutions of 2 km or finer, for both FCMP and PMP, the loss at year 200 is about 3500–3600 Gt. The losses as a function of resolution are similar for $p = 0$ and $p = 0.5$, with losses at $p = 0.5$ always greater than those at $p = 0$.

Convergence of results as a function of resolution is slower with NMP (no melt in partly grounded cells) than with FCMP or PMP. With NMP, the IMAF loss at 1 km is $\approx$200–300 Gt less than at 0.5 km. Even at 0.5 km, the loss is about 300 Gt less than for FCMP and PMP, suggesting that the results have not yet converged. This behavior is in contrast to Seroussi and Morlighem (2018), who found low sensitivity to resolution when using a no-melt parameterization in the Ice Sheet System Model (ISSM).

**Table 4.** Loss in IMAF (Gt) at the end of experiment Ice1rr for all values of $p$ and melt rate parameterizations. The loss in total ice mass (Gt) is shown in parentheses.

| | Resolution | Melt parameterization | | |
| --- | --- | --- | --- | --- |
| | | FCMP | PMP | NMP |
| p=0 | 8 km | 3243 (6508) | 3188 (6442) | 2330 (5156) |
| | 4 km | 3297 (6815) | 3236 (6722) | 2446 (5566) |
| | 2 km | 3470 (7099) | 3417 (7025) | 2636 (5915) |
| | 1 km | 3598 (7287) | 3547 (7216) | 3003 (6451) |
| | 0.5 km | 3553 (7245) | 3527 (7209) | 3223 (6782) |
| p=0.5 | 8 km | 3255 (6521) | 3197 (6451) | 2337 (5168) |
| | 4 km | 3394 (6942) | 3342 (6860) | 2539 (5689) |
| | 2 km | 3552 (7214) | 3499 (7137) | 2721 (6032) |
| | 1 km | 3630 (7339) | 3581 (7272) | 3065 (6544) |
| | 0.5 km | 3603 (7337) | 3579 (7304) | 3318 (6936) |
| p=1 | 8 km | 3112 (6392) | 3084 (6347) | 2201 (5065) |
| | 4 km | 2961 (6537) | 2936 (6495) | 2381 (5648) |
| | 2 km | 3891 (7840) | 3869 (7809) | 3491 (7241) |
| | 1 km | 4276 (8382) | 4263 (8363) | 4024 (8007) |
| | 0.5 km | 4454 (8659) | 4448 (8650) | 4335 (8481) |

Table 5 shows the loss of ice mass (including grounded ice) after 100 years for Ice1r with $p = 0$ and $p = 1$, melt parameterizations PMP and NMP, and as a function of resolution. These results can be compared directly with Tab. 3 in Seroussi and Morlighem (2018) at resolutions of 0.5, 1, and 2 km. Their melt parameterization SEM1 is similar to our PMP, and the Weertman and Tsai friction laws correspond closely to $p = 0$ and $p = 1$, respectively. The caption of Tab. 3 in Seroussi and Morlighem (2018) should read "Change in ice mass (in Gt)" instead of "Change in volume above flotation ($\Delta$ VAF in Gt)" (H. Seroussi, personal communication, August 11, 2020).

With $p = 0$ and using NMP, our results are within 10% of those in Seroussi and Morlighem (2018) at corresponding resolutions. For CISM, NMP mass loss is still increasing with finer resolution at 0.5 km, while their NMP results show little sensitivity to resolution. Their results with SEM1 depend strongly on resolution and have mass loss decreasing with finer resolution, while our results with PMP depend much more weakly on resolution and have mass loss increasing with resolution. Their Weertman results using SEM1 at a resolution of 0.125 km are close ($\sim 1\%$) to our $p = 0$ results using PMP at 0.5 km.

CISM results with $p = 1$ and NMP are within 10% of ISSM results with a Tsai law and NMP. In this case, mass losses increase with finer resolution for both models. With $p = 1$ and PMP, the CISM mass loss increases with finer resolution and is close ($\sim 2\%$) to the NMP value at 0.5 km. For ISSM with the Tsai law and SEM1, however, the mass loss at corresponding

**Table 5.** Loss in ice mass (Gt) at the end of experiment Ice1r for $p = 0$ and $p = 1$ and melt rate parameterizations PMP and NMP. These values can be compared directly to those in Seroussi and Morlighem (2018), Tab. 3, which are shown here in parentheses. The SEM1 and Tsai parameterizations in Seroussi and Morlighem (2018) are similar to PMP and $p = 1$, respectively, in CISM.

| | | Melt parameterization | |
| | Resolution | PMP | NMP |
|---|---|---|---|
| | 8 km | 3999 | 3289 |
| | 4 km | 4247 | 3573 |
| p=0 | 2 km | 4358 (5672) | 3787 (4132) |
| | 1 km | 4414 (5235) | 4042 (4130) |
| | 0.5 km | 4412 (4824) | 4212 (4120) |
| | 8 km | 3999 | 3301 |
| | 4 km | 4179 | 3700 |
| p=1 | 2 km | 4753 (6614) | 4448 (4943) |
| | 1 km | 5035 (6365) | 4848 (5150) |
| | 0.5 km | 5203 (6034) | 5110 (5374) |

resolutions is much greater than in CISM, and is still far apart ($\approx$800 Gt) at 0.5 km. Thus, there appear to be large differences between models in the treatment of melt near the grounding line, even for two schemes (PMP and SEM1) that are conceptually similar. The reasons for the differences are unclear, and a detailed comparison of model numerics is beyond the scope of this study.

Focusing again on CISM results, Tab. 4 and 5 and Fig. 10 show greater sensitivity to resolution and (at higher resolution) greater IMAF losses with $p = 1$ than with $p = 0$ and 0.5. The high-resolution results show that the grounding line retreats faster when the bed is more lubricated, which is consistent with other MISMIP+ models (Cornford et al., 2020). After 100 and 200 years, IMAF losses with $p = 1$ at 1 km are within 4% of the losses at 0.5 km for both FCMP and PMP, and within 7% for NMP. At the end of Ice1rr, the loss at 0.5-km resolution is about 4400 Gt for each melt scheme. At coarser resolutions, however,

NMP has consistently lower IMAF losses than FCMP and PMP. The coarser the resolution, the greater the difference: about 11%, 23%, and 40% at 2, 4, and 8 km, respectively.

For a given melt parameterization, we might expect that increasing the lubrication at the bed should lead to faster flow towards open water and greater IMAF loss in response to basal melting. Tab. 4 and 5 show this to be true for resolutions of 2 km and higher. For resolutions of 4 and 8 km, however, IMAF losses with $p = 1$ are less than those for $p \leq 0.5$, indicating larger

error at coarse resolutions for $p = 1$. Since the flow factor $A$ is tuned in the Stnd experiment to adjust the initial grounding-line location, differences in both viscosity and bed lubrication can influence the transient response.

During experiment Ice1ra, for all $p$ and melt parameterizations, the grounding line continues retreating, despite the removal of basal melt. This means that the grounding line is still adjusting to the strong melting of the first 100 years (in contrast to

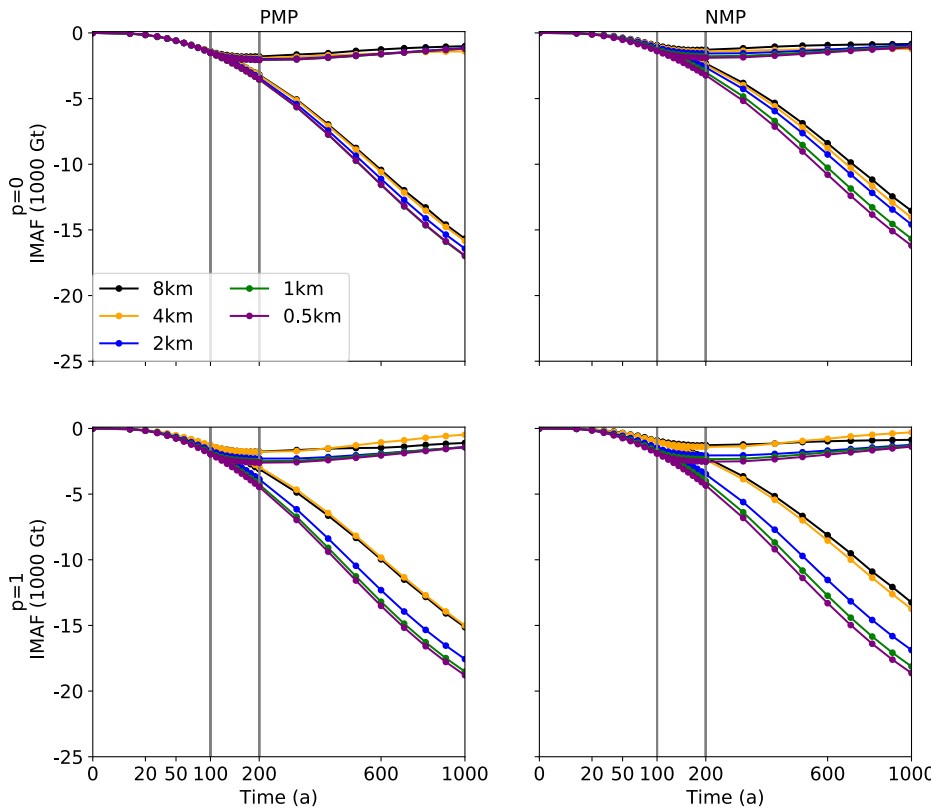

**Figure 11.** Evolution of IMAF relative to steady state for the full set of moderate-melt experiments. The layout is similar to Fig. 10, but showing results only for $p = 0$, $p = 1$, and melt parameterizations PMP and NMP. The vertical line at time 100 a indicates the end of experiment Ice1r and the start of Ice1ra/Ice1rr. The vertical line at 200 a shows the end of Ice1ra/Ice1rr and the start of Ice1rax/Ice1rrx.

MISMIP3d, in which the grounding line advances as soon as the basal friction perturbation is removed). To illustrate grounding-line behavior over a longer period, Fig. 11 shows the 1000-year results including Ice1rax and Ice1rrx. Results are shown only for $p = 0$ and 1, and for PMP and NMP, since the differences between FCMP and PMP, and between $p = 0$ and 0.5, are small.

The qualitative behavior as a function of $p$ and resolution is similar to the shorter experiments, Ice1rr and Ice1ra. During Ice1rax, for all parameterizations and resolutions, the ice stream still has not returned to its original, steady-state position after 900 years without melting. This suggests that the melting is strong enough to remove ice mass on a faster time scale than it can be restored through surface accumulation. Also, in contrast to the MISMIP3d results in Sec.3.3, the presence of buttressing slows down the return to the original equilibrium.

When comparing results at 0.5 km between $p = 0$ and $p = 1$, most of the difference in IMAF occurs during the first 200 years. At that time, the IMAF loss with $p = 1$ is about 2500 Gt greater than with $p = 0$. This difference remains nearly unchanged during the 800 years of Ice1rrx. This may be in part a function of bed geometry. The grounding line originally lies on the

retrograde part of the bed, but reaches the part of the bed sloping downward ($x < 390$ km) within the first 160 to 300 years (depending on melt parameterization and resolution), after which grounding-line retreat may be less sensitive to bed lubrication. Simulations at higher resolution or with FCMP/PMP reach the downsloping part of the bed earlier than simulations at lower resolution or with NMP.

Overall, the moderate-melt experiments show that it can be beneficial to allow some basal melt in cells containing the grounding line. At resolutions of 1 km or coarser, the FCMP and PMP schemes, which permit melt in partly grounded grid cells, are more accurate (i.e., closer to the results at 0.5 km) than NMP. This difference is most striking when $p \leq 0.5$. In particular, using FCMP or PMP with $p = 0$ or 0.5 gives reasonable accuracy for transient IMAF loss at 4 km, within $\sim$5%-10% of the values at 0.5 km. At 2 km, the IMAF losses are within $\sim$3% of the high-resolution values. Using NMP, the results converge much more slowly with resolution.

These experiments also illustrate the sensitivity of grounding-line behavior to the basal sliding law, for the case of a retrograde bed with buttressing. For a well-lubricated bed ($p = 1$), resolution of $\approx$1 km is needed to obtain convergent results in steady-state and transient experiments. With $p = 0$ or 0.5, resolution of $\approx$2–4 km is sufficient.

### 4.3 High basal melt rate experiments

The moderate-melt experiments show that it is beneficial in CISM to allow some melt in cells containing the grounding line. Next, we investigate whether this is true for a high basal melt rate. We run the same tests as in Sect. 4.2, but with the melt rate from Eq. (18) multiplied by 5, such that the highest melt rate is 150 m a$^{-1}$. As for the moderate-melt experiments, results with $p = 0.5$ are similar to those with $p = 0$, and results using FCMP are similar to those using PMP. Therefore, we show results only for $p = 0$ and 1 and melt parameterizations PMP and NMP. We use the same protocols and spin-ups as in Sect. 4.2.

Figure 12 shows results from the full set of high-melt experiments. Compared to the moderate-melt experiments, the results at 8 km are more distinct from those at 4 km for all $p$-values and melt parameterizations. IMAF loss is always smaller at coarser resolution. At year 200, with $p = 0$ and PMP, results at 4-km resolution are within 10% of the IMAF changes at 0.5 km, and results at 1–2 km are within 3%. In contrast, when $p = 1$ (for both PMP and NMP), and when $p = 0$ with NMP, resolution of 1 km is needed to obtain IMAF loss within 10% of the value at 0.5 km. Differences are up to 20% at 2 km, and more than 35% at 4 km or coarser. These results show that with a higher melt rate, convergence to the high-resolution solution is slower for $p = 1$ and NMP. As with moderate melt, convergence is better with PMP than with NMP for all $p$ values.

### 4.4 Calving experiments

The next set of experiments is based on the MISMIP+ Ice2 experiments (Asay-Davis et al., 2016), which apply a high melt rate in the ice-shelf region to imitate a major calving event:

$$
m = \begin{cases} 0 \text{ m a}^{-1}, & x < 480 \text{ km}, \\ 100 \text{ m a}^{-1}, & x \geq 480 \text{ km}, \end{cases} \tag{19}
$$

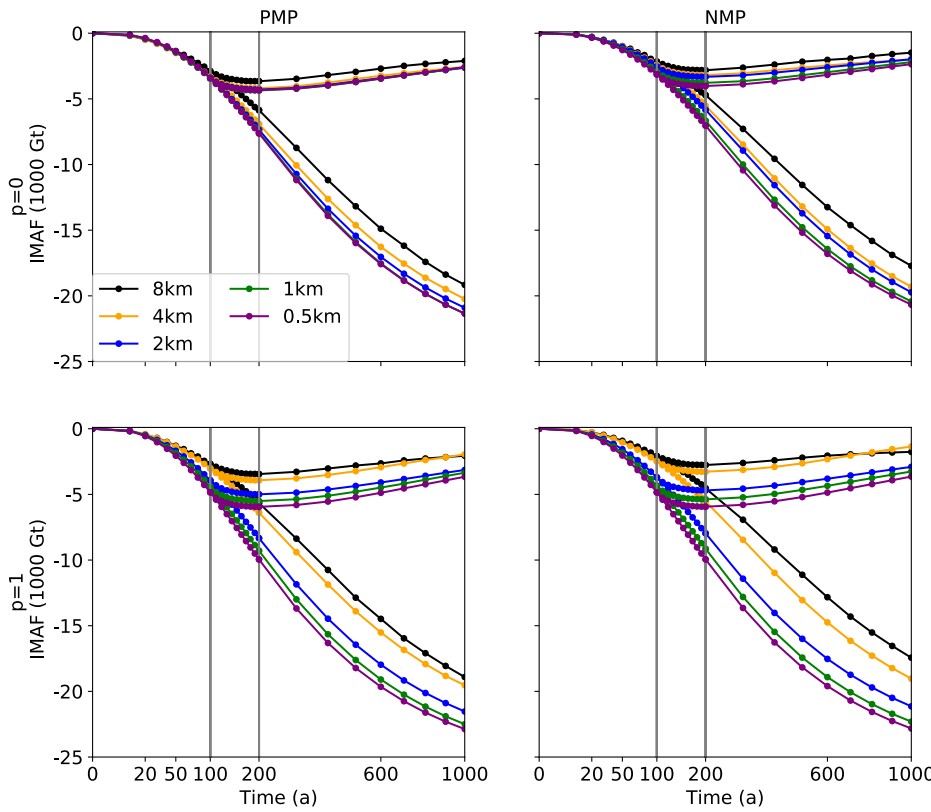

**Figure 12.** Evolution of IMAF relative to steady state for the full set of high-melt experiments. The layout is the same as in Fig. 11.

These experiments test the model's ability to reach a new stable grounding line on a retrograde bed, as well as the sensitivity to the treatment of melt in partly grounded cells. We run experiments Ice2rr and Ice2ra to year 200, followed by Ice2rrx and Ice2rax to year 1000. Protocols are the same as for the Ice1 set of experiments in Sects. 4.2 and 4.3, except for the modified melt rate. As for the previous experiments, the results with $p = 0.5$ are similar to those with $p = 0$, and results using FCMP are similar to those using PMP. Therefore, we show results only for $p = 0$ and 1 and melt parameterizations PMP and NMP. These results can be compared to the multi-model Ice2 experiments in Cornford et al. (2020).

Fig. 13 shows the evolution of IMAF for the full set of experiments. Convergence as a function of resolution is similar with PMP and NMP, in contrast to the previous moderate- and high-melt experiments (Sects. 4.2 and 4.3), which showed faster convergence for PMP. For both melt schemes and both $p$ values, the results at 2 km are close to those at higher resolution. At year 1000 of the retreat experiment (Ice2rrx), the IMAF changes at 2-km resolution are within 2% and 5% of the 0.5-km values for $p = 0$ and $p = 1$, respectively. The errors are larger at 4 km and 8 km, reaching several hundred Gt at year 1000 for most experiments. For PMP with $p = 0$, IMAF stagnates in the Ice2rax experiments at 4 km and 8 km, while steadily recovering at resolutions of 2 km or finer.

Figure 14 shows the grounding-line evolution over 1000 years for $p = 1$. Results with $p = 0$, not shown, are similar. We show results at resolutions of 2 km, where the solutions are well converged, and 8 km, where they are far from converged. The left-hand and right-hand plots show the grounding-line location in midchannel and at the domain edge, respectively. The grounding line is exposed to large melt rates only at the domain edge, where it extends farther downstream. In midchannel, the grounding line retreats rapidly for the first 100 years for both melt schemes and resolutions; it then keeps retreating for Ice2rr and steadily recovers for Ice2ra. At the domain edge, however, the qualitative behavior varies with resolution. At 2 km, for both melt schemes, the grounding line holds steady near its initial location in the Ice2rr/Ice2rrx experiments, while decreasing slightly (by about 3 km) after year 200 in Ice2ra/Ice2rax before stabilizing. At 8 km, the grounding line holds steady only in the 1000-year retreat experiments with NMP. For NMP with readvance and for both PMP experiments, the grounding line retreats signifcantly (by 5–15 km) at the domain edge after year 100.

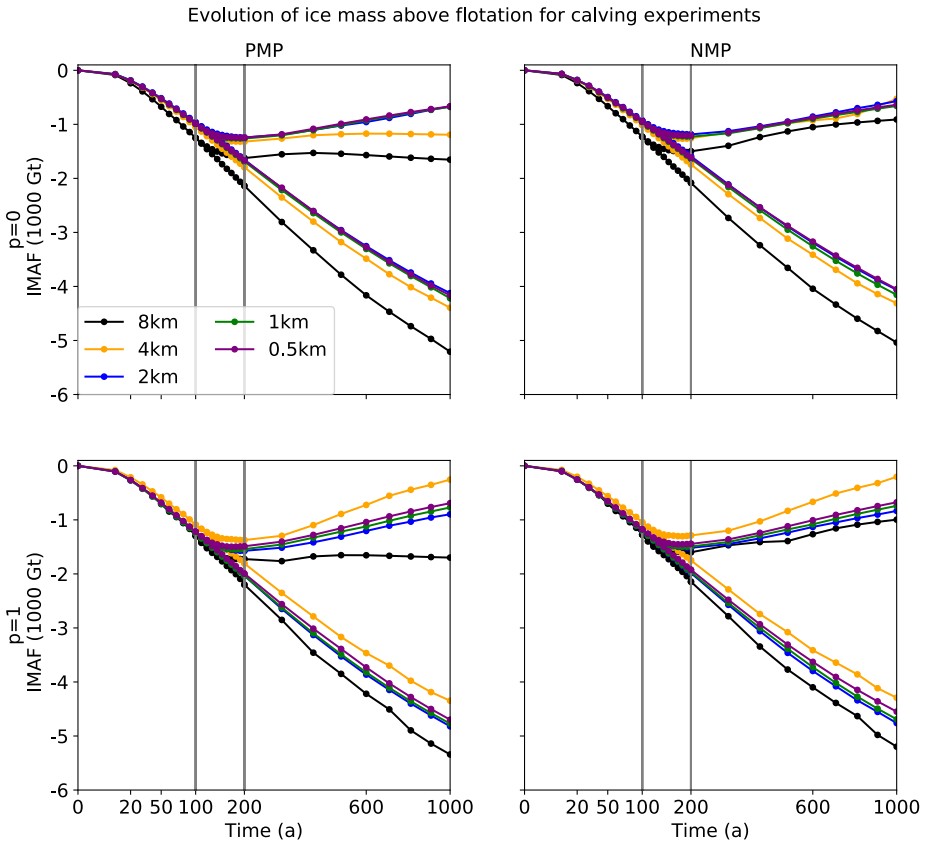

**Figure 13.** Evolution of IMAF relative to steady state for the full set of calving experiments. The layout is the same as in Fig. 11, but for Ice2 instead of Ice1 protocols.

This figure can be compared to Fig. 13 (for midchannel) and Fig. 14 (for the domain edge) in Cornford et al. (2020), which show the mean grounding-line location and its spread for multiple models over the first 200 years. These authors found that

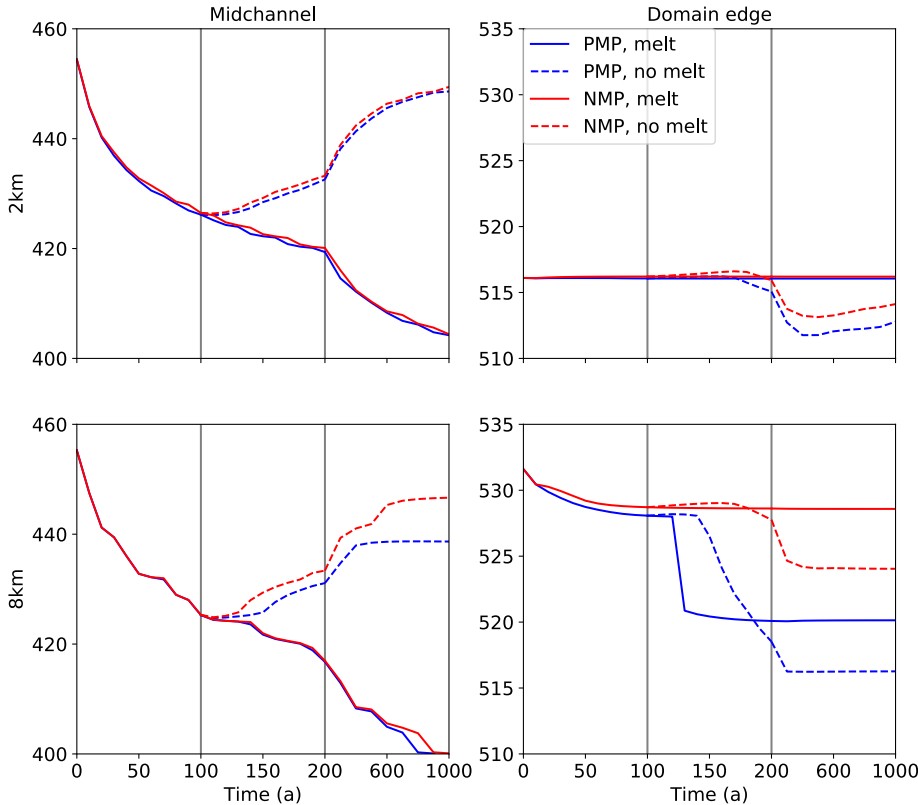

**Figure 14.** Evolution of the grounding-line position (km) for calving experiments. We show two resolutions, 2 km (top) and 8 km (bottom), for the midchannel (left) and the domain edge (right), and for the PMP (red) and NMP (blue) melt parameterizations. Solid lines denote continuous retreat (Ice2r/Ice2rr/Ice2rrx), and dashed lines denote recovery following the first 100 years of retreat (Ice2ra/Ice2rax). The time axis is scaled to focus on the first 200 years. Vertical lines are as in Fig. 11.

the domain-edge results could be divided into two groups. The models in group A, which do not use a subgrid interpolation scheme to compute melt rates (similar to CISM's NMP), show substantial retreat along the domain edge. This is in contrast to models in group B, which do use such a scheme (like CISM's PMP), or otherwise have a special treatment of calving fronts. The group B models show substantial grounding-line retreat at the domain edge, starting early in the run. In midchannel, there is little difference between the two groups.

At resolutions of 2 km or finer, CISM's PMP and NMP runs both behave like group A, the models without subgrid interpolation of melt rates. Only at coarser resolution do some CISM experiments show group B behavior, with substantial retreat at the domain edge. This result suggests that compared to subgrid interpolation schemes in other models, CISM's PMP scheme is less prone to grounding-line retreat in the presence of large melt rates.

**Table 6.** Ice softness $A$ ($10^{-18}\mathrm{Pa}^{-3}\mathrm{a}^{-1}$) and resulting grounding line position $x_g$ (km) at the end of the spin-up for the slow-moving ice shelf experiment. Note that the $A$-values are not truncated.

| Resolution | p=0 | | p=1 | |
|---|---|---|---|---|
| | A | $x_g$ | A | $x_g$ |
| 8 km | 2.5 | 452.10 | 2.05 | 455.27 |
| 4 km | 3.0 | 453.97 | 2.7 | 454.36 |
| 2 km | 3.1 | 454.68 | 2.8 | 454.42 |
| 1 km | 3.1 | 456.24 | 2.9 | 455.44 |
| 0.5 km | 3.1 | 455.86 | 2.9 | 454.33 |

## 4.5 Slow-moving ice shelf experiments

Seroussi and Morlighem (2018) suggested that applying a melt rate to partly grounded cells would lead to unrealistic retreat in configurations with slow-moving (or static) ice shelves. To test this idea, we ran a set of slow-moving shelf experiments using the same MISMIP+ setup as in Sect. 4.2, including the melt rate of Eq. (18), but with an accumulation rate of 0.05 m a$^{-1}$ instead of 0.3 m a$^{-1}$. To obtain steady-state grounding lines in the desired location, we ran a new set of spin-ups over 60,000 years, with adjusted ice softness values given in Tab. 6. The grounding lines lie within $\approx 2$ km of $x = 454$ km in all cases.

With lower accumulation, the maximum ice-shelf speed at the grounding line is $\approx 66$ m a$^{-1}$, compared to $\approx 420$ m a$^{-1}$ in the moderate-melt experiments.

Figure 15 shows the change in IMAF, which is about half as large at a given time as in the moderate-melt experiments. As in both the moderate- and high-melt experiments, the PMP scheme shows better convergence than NMP, regardless of $p$. With PMP and $p = 0$, the results vary weakly with resolution, with little difference between 8 km and 4 km. For this case, the IMAF

loss decreases with higher resolution, in contrast to the other results shown so far. With NMP and $p = 0$, the convergence is somewhat better than for the moderate- and high-melt experiments; a resolution of 2 km (arguably 4 km) is adequate to accurately quantify IMAF loss. When $p = 1$, the behavior is qualitatively similar to previous experiments, and again PMP gives better convergence than NMP. At 0.5 km, the IMAF loss with NMP is within $\approx 5.6\%$ and $\approx 1\%$ of the PMP values when $p = 0$ and $p = 1$, respectively. Thus, we do not find that slower shelf flow degrades the accuracy of PMP relative to NMP.

Figure 15 also shows that throughout experiments Ice1ra and Ice1rax, the grounding line does not even begin to re-advance, in contrast to the experiments with higher accumulation. This suggests that accumulation, rather than buttressing, sets the re-advance time scale.

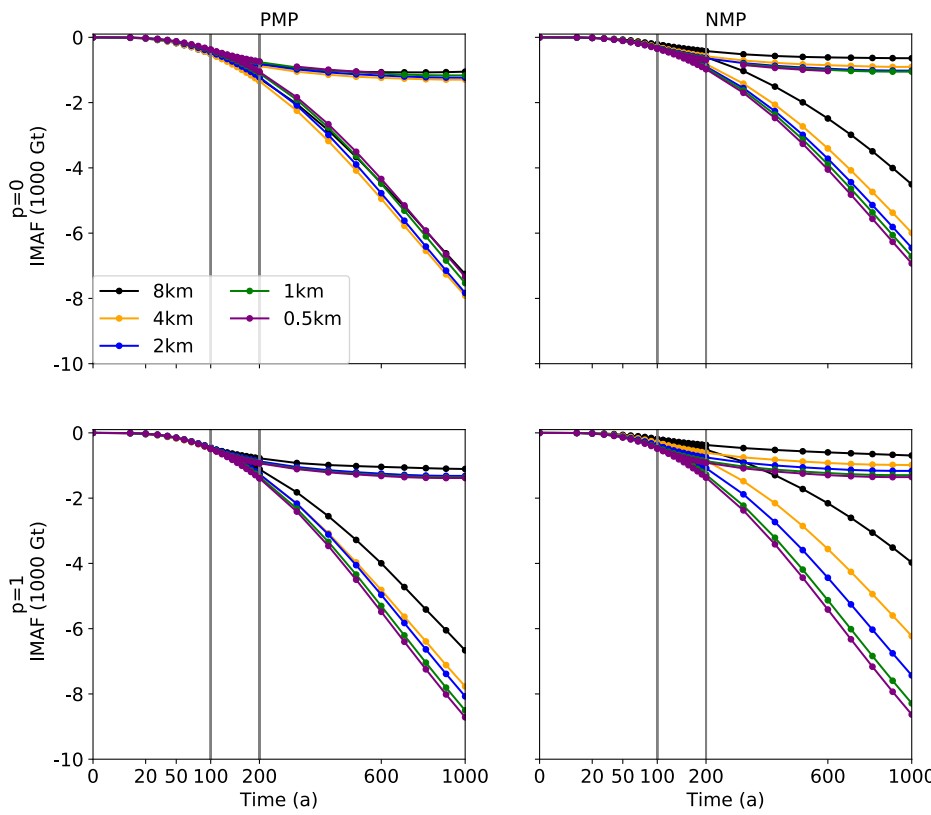

**Figure 15.** Evolution of IMAF relative to steady state for the full set of slow-moving ice shelf experiments, when the melt rate is the same as in the moderate-melt experiments (Sect. 4.2), but the shelf flows more slowly as a result of reduced accumulation. The layout is the same as in Fig. 11 (but note the different scale on the $y$ axis).

## 5 Discussion and conclusions

We have studied the effects of different ice-flow approximations, basal-friction and basal-melting parameterizations, and grid
resolution in marine ice-sheet simulations with the Community Ice Sheet Model. We used the experimental frameworks of MISMIP3d, to explore the response of an ice stream to basal friction perturbations, and MISMIP+, to test the ice-sheet sensitivity to sub-ice-shelf melting in the presence of strong lateral buttressing. Using a basal sliding law with power-law and Coulomb behavior as asymptotic limits, we varied a parameter $p$ that quantifies the connectivity of the subglacial hydrology system to the ocean; $p = 1$ implies a strong connection, while $p = 0$ implies no connection. We tested CISM's SSA, DIVA,
and BP solvers with various grid resolutions (0.5 to 8 km), connectivity parameters ($p = 0$, 0.5, and 1), and treatments of sub-shelf melting (with or without melting in partly grounded cells). For each configuration or parameter choice, we evaluated how quickly the model results converge with increasing grid resolution, to guide the choice of an appropriate resolution for large-scale simulations.

For both the MISMIP3d and the sub-ice-shelf melt experiments, and for any choice of $p$ or grid resolution, the results with the DIVA solver are close to those with the more expensive BP solver, consistent with CISM simulations of Greenland and Antarctica (Lipscomb et al., 2019, 2021a). The SSA solver, which is valid when basal stresses are low enough that there is little vertical shear, gives results similar to DIVA and BP in the MISMIP+ framework, but overestimates the ice viscosity for MISMIP3d, leading to greater grounding-line advance. Since DIVA is depth-integrated, its computational cost is comparable to that of SSA. These results suggest that DIVA provides a good compromise between accuracy and cost for marine ice sheets with significant vertical shear.

While an adequate Stokes approximation is essential for accurately simulating ice flow, the choice of basal friction law can be equally or more important. For all experiments, the results are sensitive to whether the transition from finite to zero basal friction is abrupt ($p = 0$) or gradual ($p = 1$). The grounding line is always more advanced with smaller $p$. Cornford et al. (2013) suggested that high resolution ($\sim$100 m) is necessary to adequately capture grounding-line motion. Other studies, however, have shown that this requirement can be relaxed by using a subgrid-scale parameterization of basal friction near the grounding line (Gladstone et al., 2010; Leguy et al., 2014; Seroussi et al., 2014). We find this to be the case in CISM as well. Using a basal-friction GLP greatly reduces numerical errors and improves convergence when $p \leq 0.5$, although MISMIP3d runs show little or no benefit from a GLP when $p = 1$. With a GLP, the results of all experiments at 1 km are close to those at 0.5 km. Results at resolutions of 2–4 km are generally close to the high-resolution results when $p \leq 0.5$, in the sense that the numerical errors caused by coarse resolution are small compared to uncertainties associated with ice physics, including basal friction. When $p = 1$, grid resolution of 1 km is necessary for a good representation of the reference results at 0.5 km in the MISMIP3d framework, compared to 1–2 km in the melt experiment framework.

The choice of basal friction law remains a source of great uncertainty in ice sheet models (Schmeltz et al., 2002; Gillet-Chaulet et al., 2016; Brondex et al., 2017, 2019). It is difficult to evaluate simple parameterizations without comparing to more realistic models of subglacial hydrology, an active area of research (De Fleurian et al., 2018). The best choice of $p$ cannot be deduced a priori, but depends on the physics of a specific ice-stream system. Joughin et al. (2019) showed that a sliding law with $p = 1$ is the best fit for Pine Island Glacier, but more studies are needed to know if this choice is best for all marine ice sheets. Suggesting an appropriate value for Antarctic ice sheet modeling is beyond the scope of this paper. We can state, however, that if a lower value ($p \leq 0.5$) is applied instead of a higher value ($p = 1$), the simulated grounding-line behavior (when using a GLP) will likely be less sensitive to grid resolution, at least in settings similar to MISMIP3d and our melt experiments.

An issue that has received less attention is the treatment of sub-ice-shelf melting near the grounding line. Using ISSM, Seroussi and Morlighem (2018) found that applying no melt in cells containing the grounding line leads to faster convergence with increasing grid resolution. Along with Cornford et al. (2020), they argued against applying any melt in partly grounded cells. Our melt experiments with CISM, in configurations based on MISMIP+, support a different conclusion. For experiments with moderate melt, high melt, and a slow-moving ice shelf, we found faster convergence with grid resolution when using the FCMP and PMP schemes, which allow some melting in partly grounded grid cells, than when using NMP, which allows no melting in partly grounded cells. For calving experiments, NMP is arguably superior to PMP at low resolution (4 km or coarser), but NMP and PMP are similar at higher resolution (2 km or finer). Differences are small between FCMP (in which

basal melt is applied to cells that satisfy a flotation condition) and PMP (in which the applied basal melt is proportional to the
floating fraction in partly grounded cells). Overall, both FCMP and PMP outperform NMP.

We do not suggest that previous results are mistaken, but rather that the question of how to treat basal melt near the grounding line is complex, and deserving of closer study. Applying melt to partly grounded cells can drive unrealistic grounding-line retreat, but neglecting this melt can spuriously reduce the retreat rate, resulting in comparable or larger errors. The optimal treatment can depend on the ice-stream and ice-shelf geometry, the magnitude of the melt rate, and numerical details of the ice sheet model (including, perhaps, the grid structure and the staggering of variables). There is likely not a single best approach for all models and circumstances. It would be challenging, however, to apply different melt parameterizations as a function of geographical location or time, because this would involve runtime choices in where to apply which parameterization. Testing melt parameterizations in idealized settings such as MISMIP+ can show the impact of each approach in a particular model.

For continental-scale simulations, ice sheet models are typically run at resolutions of 4 km or coarser, although higher resolution near grounding lines is possible in models with grid refinement (Seroussi et al., 2019; Cornford et al., 2015, 2016). Models are often initialized using data assimilation or nudging to match an observational target such as ice velocity, thickness, or grounding-line location (Seroussi et al., 2019). With observational fitting, the initial state may be relatively insensitive to grid resolution, since errors due to inadequate resolution can be absorbed in tuned parameters. Nonetheless, the model's transient response can be sensitive to grid resolution. Lipscomb et al. (2021a) found moderate sensitivity to grid resolution in multi-century, ocean-forced Antarctic Ice Sheet experiments with CISM when comparing results at 2 km and 4 km. This sensitivity was less, however, than the sensitivity to sub-shelf melting and basal friction parameterizations.

Our transient MISMIP3d experiments (Figs. 6–7) suggest that, on century time scales, grid resolution of 2–4 km may be sufficient, giving an error in grounding-line location of only a few kilometers compared to a simulation at 1-km or higher resolution. Likewise, the melt rate experiments (Figs. 10–15) show that the transient response is relatively insensitive to resolution, provided the initial grounding-line location is similar. That is, the response depends strongly on local bed topography, especially when the grounding line lies on a retrograde bed. The dependence is minimized, however, if the spin-up is tuned toward a specific grounding-line location. Thus, the quality of the initial state—and in particular, the agreement of the initial grounding-line location with observations—may be more critical than grid resolution for simulating the transient response. This finding is in agreement with Rückamp et al. (2020), and suggests that grid resolution of ≈2–4 km near the grounding line, when accompanied by careful initialization and model verification, is appropriate for large-scale, transient Antarctic simulations. Finer resolution of ≈1–2 km may be needed when using basal sliding laws with Coulomb behavior or strong connectivity to the ocean.

*Author contributions.* GRL co-designed and ran all experiments, processed and analyzed the results, and co-wrote the manuscript. WHL implemented code changes in CISM, co-designed the experiments, analyzed the results, and co-wrote the manuscript. XSAD co-designed the experiments, analyzed the results, and contributed to the manuscript.

*Competing interests.* The authors declare that they have no conflict of interest.

*Code availability.* CISM is an open-source code developed on the Earth System Community Model Portal (ESCOMP) Git repository at *https://github.com/ESCOMP/CISM*. The version used to perform the simulations in this work is tagged as CISM_MELT_PARAM and has been archived at *https://doi.org/10.5281/zenodo.4784788* (Lipscomb et al., 2021b).

*Data availability.* The model initial files, output files and grounding line post-processing files for the experiments described in Sects. 3 and 4 are archived on the UCAR/NCAR DASH Repository at *https://doi.org/10.5065/k3ws-2435* (Leguy and Lipscomb, 2021).

*Acknowledgements.* We would like to thank the editor, Jan De Rydt, an anonymous reviewer, and Rupert Gladstone for constructive feedback that improved the quality of the paper. This material is based upon work supported by the National Center for Atmospheric Research, which is a major facility sponsored by the National Science Foundation under Cooperative Agreement No. 1852977. Computing and data storage
resources, including the Cheyenne supercomputer (doi:10.5065/D6RX99HX), were provided by the Computational and Information Systems Laboratory (CISL) at NCAR. Support for XSAD was provided through the Scientific Discovery through Advanced Computing (SciDAC) program funded by the U.S. Department of Energy (DOE), Office of Science, Advanced Scientific Computing Research and Biological and Environmental Research programs. GRL and WHL acknowledge the SciDAC program for supporting early development of CISM's grounding-line parameterizations.

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
