# Peer review of "Marine ice-sheet experiments with the Community Ice Sheet Model"

_The Cryosphere, 2020_

## Referee Comment (RC1) · Anonymous Referee #1 · 13 Feb 2021

I apologize for my late review.

Proper modelling of grounding line dynamics remains one of the main challenge for ice-sheet models. A suite of inter-comparison exercise has shown that models results are particularly sensitive to the mesh resolution in the vicinity of the grounding line. Sub-grid parameterisations have been shown to reduce the mesh size sensitivity allowing to give good results at resolutions that become achievable with typical grid sizes used in large scale simulations. Here the authors used two benchmark experiments (MISMIP3D and MISMIP+) to study the sensitivity of the results to sub-grid parametrisations of the basal friction and basal melting. The authors already studied the basal friction parametrisation in Leguy et al. (2014) with a flow line experiment and, here, extend this previous study to a 3D experiment. For the basal melting they describe sub-grid

scheme implemented in CISM.

Intercomparison exercices usually focus on the inter-model differences and I found always usefull to have detailed studies with individual models.

Most model report that sub-grid parametrisations decrease the model sensitivity to the grid size. In agreement with their previous 1D study, the authors found that the grid size sensitivity is decreased when there is a smooth transition of the basal friction in the GL vicinity. Results for basal melting are in contrast to previous studies that have reported that applying melt in partially grounded cells might lead to inaccurate results.

In conclusion, this in an interesting numerical study with a well established ice-sheet model. The manuscript is well written and clearly describe the experiments and results and I have only minor comments or questions detailed below.

Minor comments :

- Page 2, line 31, *depth-integrated versions of the Blatter-Pattyn approximation*: not sure if this is the good formulation, these cited models are indeed depth intergrated but they approximate the 3D Blatter-Pattyn model.

- Page 2, line 31, *The required resolution is coarser for sliding laws in which basal stresses are continuous across the grounding line*. Maybe this is not as easy and depends on the transition. See for example results from Gagliardini et al. (2016) and discussion on this subject in Galdstone et al. (2017).

- Page 2 Line 54, *but less so for models configured to solve the full Stokes flow equations*. Cheng et al. (2020) report that similar accuracy is obtained using sub-grid modeling with more than 20-times-coarser meshes in a Full-Stokes model. Please provide more references for sub-grid scheme in FS model to support this sentence.

- Page 3, Line 64 : *to obtain more accurate results*. Explain the meaning of ac-
curate in this context. Seroussi and Morlighem (2018) and Conford et al. (2020) report that sub-grid schemes can result in numerical errors not inaccuracies.

- Section 2 *model description*; I think not all readers will be familiar with the staggered grid. As this is used for both parameterisation maybe it could be benefical to have a small subjection before 2.1 to describe the CISM grid and introduce here that friction has to be computed at cell vertices and melt at cell corners.

- Section 2.2 Grounding line parametrisation for basal friction, from lines 144 to 151 ; It would be usefull to add a figure (or maybe in Fig.1) to illustrate this example.

- Page 6, line 153. Maybe start to say that you compute an *effective* basal friction coefficient using the friction law presented in 2.1 then that the sub-grid scheme is applied to ths coefficient.

- Figure 2. Maybe add the cell centers in your Figure.

- Page 7, Line 165-166 : *For buttressed ice shelves, however, the dynamics are more complex (Gudmundsson, 2013), and it is not obvious which melt treatment is best.*. Please explain what do you mean by more complex and not obvious.

- Page 8, Lines 188-192 : I don't really unsderstand this part and why this is here. In seems stange to say here that CISM usually the quadrant method but that another method has been presented before. See prvious comment ; maybe it would be benefical to have a specific section in the beginning of section 2 to explain how grounded fractions are computed at cell vertices and corners.

- Page 8, bottom line. *uniform basal shear stress factor*. C was introduced as a *coefficient* (page 4 top line) and is referred to as *shear stress factor* in the tables. Maybe better to use *basal shear stress factor, C,* eveywhere.

- Page 9, line 221 : *We will consider an experiment to be reversible if the difference in grounding-line location is 4 km or less.* Maybe give a better justification for this 4km.

- Section 3.2 ; might be beneficial to have distinct sub-sections for the setady state solution and the transient.

- Page 9 bottom line , *When the grounding-line position no longer changes significantly as resolution is increased, we consider the solution to have converged..* Please quantify *significantly*.

- Page 10,top line ; Maybe would be more clear to break this sentence in two ; and following previous comment it would be more precise with a given threshold to define the convergence. Or avoid to use *converged* if there is no given criteria.

- Page 11, Line 249 : *and far cheaper than BP.* Could you give numbers ?

- Fig. 5. For p=0 there is no difference between Stnd and P75R ?

- Table 5 ; would be usefull to direcly add the values from Seroussi and Morlighem (2018) here.

- Page 21 lines 414-415 : *For a given melt parameterization, increasing the lubrication at the bed should lead to faster flow towards open water and greater IMAF loss* . Not sure if this is as simple as the rate factor is tune so that the grounding line is at the same postion, so in steady state the fluxes through the grounding line should be the same ; and as the rate factor has been adjusted this might also change the buttressing ?

- General comment on the sub-melt scheme ; In Cornford et al. (2020), the effect of the basal melt parametrisation is discussed and shown in their high melt scenario. The difference is especially visible in the evolution of the gorunded area

and position of the grounding line on the edges of the domain where the ice is relatively thin. I would find usefull here to show the same plots and maybe to repeat their experiment 2 ; to see if the results of PMP are consistent with the results of the subgroups using sub-grid schemes in Cornford et al. (2020).

References :

– Cheng, G., Lötstedt, P., von Sydow, L., 2020. A full Stokes subgrid scheme in two dimensions for simulation of grounding line migration in ice sheets using Elmer/ICE (v8.3). Geoscientific Model Development 13, 2245–2258. https://doi.org/10.5194/gmd-13-2245-2020

– Cornford, S.L., Seroussi, H., Asay-Davis, X.S., Gudmundsson, G.H., Arthern, R., Borstad, C., Christmann, J., Dias dos Santos, T., Feldmann, J., Goldberg, D., Hoffman, M.J., Humbert, A., Kleiner, T., Leguy, G., Lipscomb, W.H., Merino, N., Durand, G., Morlighem, M., Pollard, D., Rückamp, M., Williams, C.R., Yu, H., 2020. Results of the third Marine Ice Sheet Model Intercomparison Project (MISMIP+). The Cryosphere 14, 2283–2301. https://doi.org/10.5194/tc-14-2283-2020

– Gagliardini, O., Brondex, J., Gillet-Chaulet, F., Tavard, L., Peyaud, V., Durand, G., 2016. Brief communication: Impact of mesh resolution for MISMIP and MISMIP3d experiments using Elmer/Ice. The Cryosphere 10, 307–312. https://doi.org/10.5194/tc-10-307-2016

– Gladstone, R.M., Warner, R.C., Galton-Fenzi, B.K., Gagliardini, O., Zwinger, T., Greve, R., 2017. Marine ice sheet model performance depends on basal sliding physics and sub-shelf melting. The Cryosphere 11, 319–329. https://doi.org/10.5194/tc-11-319-2017

– Seroussi, H., Morlighem, M., 2018. Representation of basal melting at the grounding line in ice flow models. The Cryosphere 12, 3085–3096. https://doi.org/10.5194/tc-12-3085-2018

---

## Referee Comment (RC2) · Rupert Gladstone (Referee) · 27 Feb 2021

Two sets of idealised experiments have been carried out using an ice sheet model CISM that can also form part of a community Earth System coupled model (CESM). The first set of experiments investigate impact of dependence on effective pressure in sliding relations and grounding line parameterisations for basal resistance on model performance, especially self-consistency (especially convergence with resolution). These experiments have some novel aspects but generally confirm expected results, and provide a useful reference point for future use of CISM (and CESM). The second set of experiments investigates the impact of parameterising basal melt near grounding lines, again on self-consistent model behaviour. In these experiments an unexpected result is obtained: parameterisations that allow some melt in partially

grounded cells give better convergence than parameterisations that do not allow this. This has relevance not only to future use of CISM (and CESM) but potentially also to other ice sheet models and to the development of methodologies to handle high basal melt rates near to grounding lines. The paper also compares different approximations to the Stokes equations and provides some evidence to support use of a vertically integrated model (i.e. a 2D problem is solved) that is only marginally more computationally expensive that the commonly used "Shelfy Stream Approximation".

The paper is for the most part clearly written and thorough. I have only a few minor suggestions.

This is paper might have worked well as two separate papers or as a two-part paper. The sets of experiments in sections 3 and 4 are quite distinct from each other and the paper is rather long.

In a few places the authors refer to simulations that have been "spun up" or "spun up for XXX years" to steady state. But it is not clear to me how steady state is determined. Are specific criteria used to determine steady state? Please state how you determine steady state.

The experiment naming in section 4 is counter-intuitive. Your "experiment 1" comes after you've already presented a bunch of experiments based on the MISMIP3D setup, in which you used the MISMIP3D naming convention. You also seem to use the MISMIP+ naming convention within "experiment 1", which suggests that "experiment 2" refers to a bunch of experiments. So "experiment 1" is not the first experiment, it isn't even the first set of experiments, but perhaps it refers to the second set of experiments... Please make this more reader-friendly somehow. My suggestion is to call this section something like "Moderate basal melt rate experiments", and within it refer to MISMIP+ experiment names. If you need to you can add to these names to distinguish unique aspects of your simulations that are not defined by MISMIP+ naming conventions. This is just a suggestion, deal with this however seems best to you, but I hope you can see

why I don't find the current "Experiment 1" and "Experiment 2" naming helpful to the reader.

I find myself bringing up my own papers in my comments. Please don't feel under any obligation to cite these – it should go without saying that my review outcome is not conditional on you citing my papers! I just mention them as they spring to mind as relevant to specific topics.

Line by line comments follow.

Line 46 and line 52. The suggestion of 1km resolution is not a uniformly applicable result. The required resolution is dependent on several factors. Can you reword these instances to clarify this? This is also relevant: https://doi.org/10.3189/2012AoG60A148

Line 61. Does the 1km here refer to with or without the GLP? This needs to be clarified.

Lines 64-65. This is quite an important line because you are going to later argue the opposite. So please add a line or two to explain why previous studies suggest that no melting should be applied in the grid cell containing the GL. If I remember right Cornford found that the convergence is much worse, with a tendency to grossly overestimate GL retreat.

Line 162-163. But isn't this theory specifically for Schoof's setup which involved SSA? I think that in a model in which the stress distribution is vertically resolved the precise vertical stress distribution at the grounding line would have some dependence on the shape of the shelf downstream. It might only be a small difference, but I think it worth noting that Schoof's result her isn't supposed to apply to real ice shelves in general, just the SSA setup for which he derived a solution.

Figure 2. The GL is drawn differently in the left plot compared to the middle and right plots. I think this is probably accidental, but it is confusing to the reader. Please make them look more similar. I note that there seem to be some minor alignment issues with the boxes with each figure, but these are small enough not to be much of a distraction.

Lines 183 – 184. Needs rephrasing. Either this has been demonstrated elsewhere, in which case reference it, or else you actually have considered it and decided it is not worth reporting on in this paper, in which case please change "do not consider" to something like "we carried out preliminary tests (not shown) . . . and found this to yield unrealistic. . .".

Line 185. Maybe "small changes" -> "sub-grid resolution changes" is more specific?

Lines 198 – 202. Seems like a slightly odd choice to present conclusions at the start of a results section. I don't have a big problem with this, leave it if you like, it just reads a bit odd.

Line 212. Sorry for not referring back to the MISMIP3D paper (lazy reviewer), but one of the most important aspects of the design here is how this steady state is approached. The GL "stickiness" problem can be characterised by the existence of a region of multiple steady states (which is what leads to irreversibility). So depending on which of these states one starts with, reversibility may or may not be shown. There is more about it here: https://tc.copernicus.org/articles/12/3605/2018/ Please add a line or two about the initial conditions and how steady state is approached.

Line 222. This 4km seems arbitrary in the absence of any knowledge about the size of the GL response to the perturbation. I presume this is much greater than 4km, so please give some indication of this! Also, knowing the grid cell size would give some

context to this. I don't think the reader has yet been given this information at this point? I'm commenting as I read through so I might find this later, but some indication her would be useful to help the reader decide whether 4km is a sensible choice.

Lines 235-236. If they're directly comparable, why not add Schoof's GL position to the plot? Just as a horizontal line?

Line 339. "100-m" -> "100m thick "

Figures 9 – 12. The different lines are hard to distinguish. I think this is partly because the circles are a bit too large given how closes the lines are, and the overlie each other quite often. It is also partly because the connecting line segments are in black and the circles are outlined in black. In Figure 10 especially it is hard to distinguish between 8km and p5km (perhaps because the circles are smaller but still outlined in black, which dominates?). These figures need to be clearer. It might be that changing the black to the appropriate colour will fix it, but it is hard to say without actually seeing a modified version. I also find "p5" instead of 0.5 a bit counter-intuitive. It looks like something a programmer might write but I don't see why you wouldn't just use 0.5 in a paper. Minor detail: you refer to "vertical line" in Fig 9 caption but "vertical lines" in fig 10 caption. Did you mean to mention also the 100m line again in Fig 10?

Line 496. Here you say that results are sensitive to choice of p, but you don't actually state that you get better convergence with p=1. Better convergence with p=1 is expected based on past studies, and I would say your results do indeed support this. So I think you can make a stronger statement here than just saying that the results are sensitive to choice of p.

Line 507. Brondex 2018 (J. Glac) is probably the best reference here.

Line 527. There is a great mystery behind "numerical details of the ice sheet model" which I hope the authors (and the wider ice sheet modelling community) will investigate in the near future. One thought that springs to mind and is mon explicitly mentioned here is that the grid or mesh alignment may have some impact. I think it is fairly typical in MISMIP+ simulations for the central section of the grounding line to be aligned approximately across the flow. So if a non-adaptive structured grid is used (which is the case in the current study) then grounding line retreat may naturally occur a row at a time. If previous studies that found NMP to be the better scheme have used an unstructured mesh of triangular elements (ISSM) or an adaptive mesh (BISICLES), it might be worth considering that single element ungrounding may occur more easily in such setups than in the current CISM setup, and perhaps this could explain (part of) the difference? Please don't feel under any obligation to consider this half-baked speculation in the current paper, though perhaps it could be one factor to consider when delving further into this problem.

Final paragraph. I am not convinced that this quantification of adequate resolution is fully supported by your simulations. Bear in mind that readers will mostly not read the full paper in detail but will look at key figures and conclusions. This suggestion of 2-4km being in general sufficient is dependent on many factors, not all of which are fully explored here, and in particular, this paragraph refers to the MISMIP3D experiments and not to the melt experiments. When I look at Figures 9-12 I do not have confidence that a converged result has been achieved at 2-4km resolution. If you see, in the next year or two, a published future projection of the Antarctic Ice Sheet carried out at 4km resolution, and read that their justification for the choice of resolution is simply citing your paper, will you feel comfortable with that?

---

## Author Response (AR2)

**Editor comments**

I have indicated the need for technical corrections to allow you to include a Data Availability statement that fulfils the Copernicus Data Policy requirements (https://www.the-cryosphere.net/policies/data_policy.html)

We have added two references for code and data availability, along with the following text in the manuscript (lines 632-636):

*Code availability.* CISM is an open-source code developed on the Earth System Community Model Portal (ESCOMP) Git repository at https://github.com/ESCOMP/CISM. The version used to perform the simulations in this work is tagged as CISM_MELT_PARAM and has been archived at https://doi.org/10.5281/zenodo.4784788 (Lipscomb et al., 2021b).

*Data availability.* The model initial files, output files and grounding line post-processing files for the experiments described in Sects. 3 and 4 are archived on the UCAR/NCAR DASH Repository at https://doi.org/10.5065/k3ws-2435 (Leguy and Lipscomb, 2021).

In lines 165-166 on Page 7 and in your response to the reviewer #1, you say that it is 'not obvious which melt treatment is best for a buttressed ice shelf'. I think it is worth reiterating here that a necessary condition for any melt treatment is that, if applied to an unbuttressed SSA ice shelf in steady state, the GL should remain stationary.

We acknowledge the validity of the theoretical argument, but the situation is more complex in numerical models. As a community, we do not have efficient or agreed-upon rules for determining whether an ice shelf is buttressed. Even if we did, these computations are nonlocal and expensive, as discussed in Zhang et al. (2020; https://tc.copernicus.org/articles/14/3407/2020/). So we cannot draw a clear line between buttressed and unbuttressed shelves.

However, we agree that for simple unbuttressed shelves as in MISMIP3d, the GL should remain stationary. For a numerical model, we think it is too strong to require that the GL should be *exactly* stationary, but it is reasonable to expect that GL retreat will approach zero as the grid is refined.

We therefore ran additional experiments in the MISMIP3d framework, applying a uniform basal melt rate of 30 m/a to unbuttressed shelves that are initially in steady state. As expected, grounding-line movement is minimal for NMP: a few hundred meters at most. For PMP and FCMP, we found that GL movement is less than one grid cell width over 1000 years for all experiments at a resolution of 4 km or finer. We think that a small error which approaches zero in the limit of high resolution is an acceptable one. We did, however, observe spurious GL retreat over a distance of several grid cells in three experiments at 8-km resolution, suggesting that FCMP and PMP should only be applied at ~4 km or finer.

We have shown that in diverse settings which resemble real ice-shelf systems, the PMP and FCMP schemes are more accurate (or at least are not significantly less accurate) than NMP, without driving spurious grounding-line retreat. This includes both buttressed and unbuttressed ice shelves, with a range of melt rates, flow speeds, and basal friction parameters. Although we cannot prove that allowing some melt in partly grounded cells will *never* drive spurious retreat, we think such cases are likely to be very rare in realistic setups.

The new experiments for unbuttressed shelves are described in several paragraphs at the beginning of Section 4. On p. 7, we added the text "Thus, a melt scheme should not trigger grounding-line retreat when applied to an unbuttressed, steady-state shelf."

**Reviewer comments**

We thank both reviewers for taking the time to review our paper during these difficult times. We appreciate the many constructive comments, and we think the changes made in response to the comments will make the manuscript clearer and stronger.

Page and line references in the reviewer comments are to the original submitted manuscript, whereas page and line references in our replies are to the revised manuscript with tracked changes.

**Anonymous reviewer 1**

Proper modelling of grounding line dynamics remains one of the main challenges for ice-sheet models. A suite of inter-comparison exercises has shown that models results are particularly sensitive to the mesh resolution in the vicinity of the grounding line. Sub-grid parameterisations have been shown to reduce the mesh size sensitivity allowing to give good results at resolutions that become achievable with typical grid sizes used in large scale simulations. Here the authors used two benchmark experiments (MISMIP3D and MISMIP+) to study the sensitivity of the results to sub-grid parametrisations of the basal friction and basal melting. The authors already studied the basal friction parametrisation in Leguy et al. (2014) with a flow line experiment and, here, extend this previous study to a 3D experiment. For the basal melting they describe sub-grid scheme implemented in CISM.

Intercomparison exercises usually focus on the inter-model differences and I found always useful to have detailed studies with individual models.

Most models report that sub-grid parametrisations decrease the model sensitivity to the grid size. In agreement with their previous 1D study, the authors found that the grid size sensitivity is decreased when there is a smooth transition of the basal friction in the GL vicinity. Results for basal melting are in contrast to previous studies that have reported that applying melt in partially grounded cells might lead to inaccurate results.

In conclusion, this is an interesting numerical study with a well established ice-sheet model. The manuscript is well written and clearly describes the experiments and results and I have only minor comments or questions detailed below.

Minor comments :

• **Page 2**, line 31, depth-integrated versions of the Blatter-Pattyn approximation: not sure if this is the good formulation, these cited models are indeed depth integrated but they approximate the 3D Blatter-Pattyn model.

We agree that the depth-integrated models are distinct from BP. We modified the text (now on p. 2, l. 31) to read "along with depth-integrated higher-order approximations (Goldberg, 2011; Perego et al., 2012)."

• **Page 2**, line 44, The required resolution is coarser for sliding laws in which basal stresses are continuous across the grounding line. Maybe this is not as easy and depends on the transition. See for example results from Gagliardini et al. (2016) and discussion on this subject in Gladstone et al. (2017).

Thank you for pointing out these results and discussion. We modified the text (p. 2, l. 44-46) to read, "In higher-order models, the required resolution is coarser for sliding laws in which basal stresses are continuous across the grounding line (Leguy et al., 2014; Tsai et al., 2015; Gladstone et al., 2017). (This is not necessarily true for full-Stokes models, as discussed by Gagliardini et al., 2016.)"

• **Page 2** Line 54, but less so for models configured to solve the full Stokes flow equations. Cheng et al. (2020) report that similar accuracy is obtained using sub-grid modeling with more than 20-times-coarser meshes in a Full-Stokes model. Please provide more references for sub-grid scheme in FS model to support this sentence.

We rewrote this sentence as (p. 2, l. 56): "… for Stokes approximation solvers, as well as models configured to solve the full Stokes flow equations (Cheng et al. 2020)."

• **Page 3**, Line 64 : to obtain more accurate results. Explain the meaning of accurate in this context. Seroussi and Morlighem (2018) and Conford et al. (2020) report that sub-grid schemes can result in numerical errors not inaccuracies.

You are correct. We replaced "to obtain more accurate results" with "to reduce numerical errors" (p. 3, l. 66).

• Section 2 model description; I think not all readers will be familiar with the staggered grid. As this is used for both parameterisations maybe it could be beneficial to have a small subsection before 2.1 to describe the CISM grid and introduce here that friction has to be computed at cell vertices and melt at cell corners.

Thank you for the suggestion. We added a short paragraph before the start of Section 2.1, referencing Fig. 1 (which erroneously was not referenced in the original manuscript) (p. 3-4, l. 92-95):

"The CISM grid is shown schematically in Fig. 1. Scalars such as ice thickness H and bed topography b are located at grid cell centers, with ice velocity components (u,v) at cell vertices. Since basal melt rates modify H, they lie at cell centers, whereas basal friction is a forcing term for velocity and is defined at vertices. This staggering of variables is incorporated in the GLPs for friction and sub-shelf melting, as discussed below."

• Section 2.2 Grounding line parameterisation for basal friction, from lines 144 to 151; It would be useful to add a figure (or maybe in Fig.1) to illustrate this example.

We added a figure (the new Fig. 2) as suggested, with an explanatory caption. (p. 6)

• **Page 6**, line 153. Maybe start to say that you compute an effective basal friction coefficient using the friction law presented in 2.1 then that the sub-grid scheme is applied to this coefficient.

We added a sentence giving the functional form of beta (p. 7, l. 169) for the Weertman and Coulomb laws presented in 2.1. With this addition, we think the text is clearer on how the subgrid scheme is used to modify the coefficient beta.

• Figure 2. Maybe add the cell centers in your Figure.

We added the cell centers in this figure (now Fig. 3, p. 8). We also added a sentence in the caption stating that for the FCMP scheme, the flotation condition is evaluated at cell centers.

• **Page 7**, Line 165-166 : For buttressed ice shelves, however, the dynamics are more complex (Gudmundsson, 2013), and it is not obvious which melt treatment is best. Please explain what do you mean by more complex and not obvious.

Buttressed ice shelves impact grounding line migration via the backstress they impose on upstream grounded ice, as a result of which the GL can be stable on reverse-sloping beds. This is dynamically more complex compared to unbuttressed ice shelves, which are unstable on reverse-sloping beds. We expanded the text as follows (p. 7, l. 181-185):

"On a retrograde bed (i.e., a bed that slopes upward in the direction of ice flow), grounding lines are unconditionally unstable, assuming no flow variation in the transverse direction. Buttressing, however, can stabilize the grounding line on retrograde beds (Gudmundsson, 2013). Thus, it is not obvious which melt treatment is best for a buttressed ice shelf: a scheme that applies no melt in cells containing the grounding line, or a scheme that allows some melt in partly grounded cells."

• **Page 8**, Lines 188-192 : I don't really understand this part and why this is here. It seems strange to say here that CISM usually uses the quadrant method but that another method has been presented before. See previous comment ; maybe it would be beneficial to have a specific section in the beginning of section 2 to explain how grounded fractions are computed at cell vertices and corners.

We agree that this presentation is confusing. We tried to clarify the presentation by moving this paragraph (p. 7, l. 161-165), with some modifications, to Section 2.2, just after the equations for computing phi_g^v. The revised paragraph states that we can use the same equations on a quadrant-by-quadrant basis instead of for an entire staggered grid cell, and the reason to do this is to obtain consistent areas when summing over the staggered grid (where the area fraction is phi_g^v) and the unstaggered grid (where the area fraction is phi_g^c).

• **Page 8**, bottom line. uniform basal shear stress factor. C was introduced as a coefficient (page 4 top line) and is referred to as shear stress factor in the tables. Maybe better to use basal shear stress factor, C, everywhere.

Yes, thanks for spotting this inconsistency. We changed the text (p. 4, l. 99) to read "C is the basal shear stress factor."

• **Page 9**, line 221 : We will consider an experiment to be reversible if the difference in grounding-line location is 4 km or less. Maybe give a better justification for this 4km.

Indeed, this value is somewhat arbitrary. We thought that this was a reasonable threshold for an experiment during which the grounding line moves between 20-60 km depending on p, resolution and Stokes approximation.  To be sure that the results are not sensitive to this specific value, we repeated the analysis with smaller thresholds.  This adds two more irreversible cases at 2 km, and a third case at 1 km.

We added the following text in the discussion of the old Fig. 4 (now Fig. 5) (p. 13, l. 300-302): "Since the 4-km threshold for reversibility is somewhat arbitrary, we note that the results are not very sensitive to this threshold.  With a 2-km threshold, the SSA and DIVA tests with p = 0 are labeled as irreversible at 4-km resolution.  When the threshold is reduced to 1 km, the BP test with p = 0  at 2-km resolution becomes irreversible.  Otherwise, Fig. 5 is unchanged."

At p. 10, l. 248, we deleted the sentence, "We will consider an experiment to be reversible if the difference in grounding-line location is 4~km or less," since we have not yet mentioned grid resolution at this point in the discussion.

• Section 3.2 ; might be beneficial to have distinct sub-sections for the steady state solution and the transient.

We divided Section 3.2 into two sub-sections, as suggested.

• **Page 9** bottom line , When the grounding-line position no longer changes significantly as resolution is increased, we consider the solution to have converged. Please quantify "significantly".

We originally had in mind agreement within ~20 km of the solution from the highest available resolution (0.5 km).  However, this value is arbitrary.  We decided not to give a threshold, but instead to reword the discussion in terms of relative rates of convergence.  The text now reads (p. 11, l. 259-263):

"The threshold for 'significantly' depends on the application, but without stating a specific threshold, we can see that for p <= 0.5, the runs with a GLP converge much faster than those without a GLP.  Using DIVA with a GLP, for example, the 1-km solution differs from the 0.5-km solution by 6 km with p = 0 and by 9 km with p = 0.5 when we use a GLP.  Without a GLP, the respective differences are 37 km and 28 km.  For p = 1, however, a GLP does not clearly improve convergence."

• Page 10, top line ; Maybe would be more clear to break this sentence in two; and following previous comment it would be more precise with a given threshold to define the convergence. Or avoid to use converged if there is no given criteria.

Please see the text changes in the previous comment.

• **Page 11**, Line 249 : and far cheaper than BP. Could you give numbers ?

For these simulations, DIVA is computationally 10–40 times faster than BP depending on resolution, as mentioned on now page 15, line 335. We moved the statement to p. 12, l. 280.

• Fig. 5. For p=0 there is no difference between Stnd and P75R ?

Yes, the difference is too small to be seen in the figure (now Fig. 6). We added the following statement in the caption: "The Stnd and P75R positions are visually indistinguishable for p = 0 and p = 1."

• Table 5 ; would be useful to directly add the values from Seroussi and Morlighem (2018) here.

We added the values of Seroussi and Morlighem (2018) to the table in parentheses, and modified the caption accordingly.

• **Page 21**, lines 414-415 : "For a given melt parameterization, increasing the lubrication at the bed should lead to faster flow towards open water and greater IMAF loss." Not sure if this is as simple as the rate factor is tuned so that the grounding line is at the same position, so in steady state the fluxes through the grounding line should be the same; and as the rate factor has been adjusted this might also change the buttressing ?

Yes, thanks for catching this. At steady state, the fluxes through the grounding line should be very similar (not exactly the same due to the small differences in grounding-line positions). Modifying the rate factor changes the buttressing, with a smaller rate factor leading to more viscous ice and larger buttressing. The increased viscosity required to advance the grounding line to the desired position can offset the effects of increased lubrication.

To acknowledge this complication, we added some text (p.20, l. 425): "On the other hand, the smaller values of A (i.e., greater viscosity) for p = 1 can influence the transient response, and therefore the different responses with p = 1 relative to p <= 0.5 must be attributed to differences in both bed lubrication and viscosity."

At p. 23, l. 480, we added the following caveat: "Since the flow factor A is tuned in the Stnd experiment to adjust the initial grounding-line location, differences in both viscosity and bed lubrication can influence the transient response."

• General comment on the sub-melt scheme; In Cornford et al. (2020), the effect of the basal melt parametrisation is discussed and shown in their high melt scenario. The difference is especially visible in the evolution of the grounded area and position of the grounding line on the edges of the domain where the ice is relatively thin. I would find useful here to show the same plots and maybe to repeat their experiment 2; to see if the results of PMP are consistent with the results of the subgroups using sub-grid schemes in Cornford et al. (2020).

Now, in Sec. 4.4, we repeated experiment 2 from Cornford et al. (2020) and added a new subsection. In the revised manuscript, this section will go between the current sections 4.3 and 4.4, as it starts from the same steady states as in the moderate and high basal melt rate experiments. This section includes two new figures. One figure shows the change in ice mass

above flotation, similar to figures in the other subsections. The second figure shows the grounding line location at the edge and center of the domain at 2-km and 8-km resolution for p = 1, with PMP and NMP, for 1000 years. This figure can be compared to Figs. 13 and 14 in Cornford et al. (2020).

We are not copying the entire section here, but we can highlight the main points:
- As is the case for other experiments, results with FCMP are similar to those with PMP, and results with p = 0.5 are similar to those with p = 0.
- For resolutions of 2 km or finer, the change in IMAF is relatively insensitive to resolution for all p values and melt schemes. That is, the 2-km results are close to the 0.5-km results.
- For high resolution (2 km or finer): At the edge of the domain, the grounding line maintains a nearly constant position under Ice2rr. Under Ice2ra, the groundling line readvances slightly (by ~2 km) after the melt is turned off. This is the case for both melt schemes and p values.
- For coarse resolution (4 km or 8 km): Results are less clean. In some cases, there is grounding line retreat at the domain edge.
- Our results differ from what might be expected based on Cornford et al. (2020). In the Cornford paper, models using a subgrid melt parameterization (group B) show a grounding line retreat of ~10 km during the melt experiments, with little to no advance when the melt is removed. Our PMP results, however, are similar to the models that do *not* apply a melt parameterization (group A). At high resolution (2 km or finer), our PMP results are similar to NMP.

These results suggest that compared to other models with subgrid interpolation of basal melt rates, CISM with PMP is less prone to GL retreat under large melt rates.

References :

– Cheng, G., Lötstedt, P., von Sydow, L., 2020. A full Stokes subgrid scheme in two dimensions for simulation of grounding line migration in ice sheets using Elmer/ICE (v8.3). Geoscientific Model Development 13, 2245–2258. https://doi.org/10.5194/gmd-13-2245-2020

– Cornford, S.L., Seroussi, H., Asay-Davis, X.S., Gudmundsson, G.H., Arth- ern, R., Borstad, C., Christmann, J., Dias dos Santos, T., Feldmann, J., Goldberg, D., Hoffman, M.J., Humbert, A., Kleiner, T., Leguy, G., Lip- scomb, W.H., Merino, N., Durand, G., Morlighem, M., Pollard, D., Rückamp, M., Williams, C.R., Yu, H., 2020. Results of the third Marine Ice Sheet Model Intercomparison Project (MISMIP+). The Cryosphere 14, 2283– 2301. https://doi.org/10.5194/tc-14-2283-2020

– Gagliardini, O., Brondex, J., Gillet-Chaulet, F., Tavard, L., Peyaud, V., Du- rand, G., 2016. Brief communication: Impact of mesh resolution for MISMIP and MISMIP3d experiments using Elmer/Ice. The Cryosphere 10, 307–312. https://doi.org/10.5194/tc-10-307-2016

– Gladstone, R.M., Warner, R.C., Galton-Fenzi, B.K., Gagliardini, O., Zwinger, T., Greve, R., 2017. Marine ice sheet model performance depends on basal sliding physics and sub-shelf melting. The Cryosphere 11, 319–329. https://doi.org/10.5194/tc-11-319-2017

– Seroussi, H., Morlighem, M., 2018. Representation of basal melting at the grounding line in ice flow models. The Cryosphere 12, 3085–3096.

**Reviewer 2: Rupert Gladstone**

Two sets of idealised experiments have been carried out using an ice sheet model CISM that can also form part of a community Earth System coupled model (CESM). The first set of experiments investigate impact of dependence on effective pressure in sliding relations and grounding line parameterisations for basal resistance on model performance, especially self-consistency (especially convergence with resolution). These experiments have some novel aspects but generally confirm expected results, and provide a useful reference point for future use of CISM (and CESM). The second set of experiments investigates the impact of parameterising basal melt near grounding lines, again on self-consistent model behaviour. In these experiments an unexpected result is obtained: parameterisations that allow some melt in partially grounded cells give better convergence than parameterisations that do not allow this. This has relevance not only to future use of CISM (and CESM) but potentially also to other ice sheet models and to the development of methodologies to handle high basal melt rates near to grounding lines. The paper also compares different approximations to the Stokes equations and provides some evidence to support use of a vertically integrated model (i.e. a 2D problem is solved) that is only marginally more computationally expensive that the commonly used "Shelfy Stream Approximation".

The paper is for the most part clearly written and thorough. I have only a few minor suggestions.

This paper might have worked well as two separate papers or as a two-part paper. The sets of experiments in sections 3 and 4 are quite distinct from each other and the paper is rather long.

We thought about splitting the paper into two parts. After consideration, we decided that there would be too much repetition between the two papers, and that a paper focused only on the MISMIP3D experiments would be rather short and not very novel.

In a few places the authors refer to simulations that have been "spun up" or "spun up for XXX years" to steady state. But it is not clear to me how steady state is determined. Are specific criteria used to determine steady state? Please state how you determine steady state.

We added some specifics on how steady-state is determined.

Now, in Sect. 3.1, line 236, we revised the text to read, "In the first step, labeled 'Stnd', the model is initialized with a uniform 500-m thick slab of ice and is spun up over 20,000 years with a uniform basal shear stress factor. At the end of the spin-up, the ice sheet has reached a steady state in which the change in grounding line location is less than $10^{-3}$ m a$^{-1}$ and the change in ice thickness at the grounding line is less than $10^{-4}$ m a$^{-1}$."

In Sect. 4.1, line 400, we added: "The steady state is determined based on grounding-line location and ice thickness, as described in Sect. 3.1."

The experiment naming in section 4 is counter-intuitive. Your "experiment 1" comes after you've already presented a bunch of experiments based on the MISMIP3D setup, in which you used the MISMIP3D naming convention. You also seem to use the MISMIP+ naming convention within "experiment 1", which suggests that "experiment 2" refers to a bunch of experiments. So "experiment 1" is not the first experiment, it isn't even the first set of experiments, but perhaps it refers to the second set of experiments. . . Please make this more reader-friendly somehow. My suggestion is to call this section something like "Moderate basal melt rate experiments", and within it refer to MISMIP+ experiment names. If you need to, you can add to these names to distinguish unique aspects of your simulations that are not defined by MISMIP+ naming conventions. This is just a suggestion, deal with this however seems best to you, but I hope you can see why I don't find the current "Experiment 1" and "Experiment 2" naming helpful to the reader.

Thank you for this suggestion.  We agree that the current naming is not helpful.  We modified the names in Section 4 as follows:
- "Experiment 1" is replaced by "moderate basal melt experiments".
- "Experiment 2" is replaced by "high basal melt experiments".
- "Experiment 3" is replaced by "slow-moving ice shelf experiments".

In response to Reviewer 1, we added a section labeled as "calving experiments." We kept the MISMIP+ experiment names (Ice1rr, etc.), since these indicate when the melt is turned on or off, and will be familiar to many readers.

I find myself bringing up my own papers in my comments. Please don't feel under any obligation to cite these – it should go without saying that my review outcome is not conditional on you citing my papers! I just mention them as they spring to mind as relevant to specific topics.

Your papers were relevant to many parts of our study. We tried to cite them appropriately.

Line by line comments follow.

**Page 2**

Line 46 and line 52. The suggestion of 1km resolution is not a uniformly applicable result. The required resolution is dependent on several factors. Can you reword these instances to clarify this? This is also relevant: https://doi.org/10.3189/2012AoG60A148

Yes, this is a good point.  We revised the text as follows (p. 2, l. 52):

"With a GLP, a resolution of 1–2 km may be sufficient to accurately represent grounding-line motion (Gladstone et al., 2010; Seroussi et al., 2014; Leguy et al., 2014; Cornford et al., 2016), with either continuous or discontinuous basal friction.  The required resolution can depend on several factors, including basal drag, channel width, and bed topography (Gladstone et al., 2012)."

Line 61. Does the 1km here refer to with or without the GLP? This needs to be clarified.

The sentence now reads (p.3, l. 62), "With a GLP, a resolution..."

**Page 3**

Lines 64-65. This is quite an important line because you are going to later argue the opposite. So please add a line or two to explain why previous studies suggest that no melting should be applied in the grid cell containing the GL. If I remember right Cornford found that the convergence is much worse, with a tendency to grossly overestimate GL retreat.

We added a sentence (p. 3, l. 67): " A physical argument can be made that applying melt in the cell containing the grounding line will artificially drive retreat, by thinning grounded ice upstream of the grounding line."

**Page 6**

Line 162-163. But isn't this theory specifically for Schoof's setup which involved SSA? I think that in a model in which the stress distribution is vertically resolved the precise vertical stress distribution at the grounding line would have some dependence on the shape of the shelf downstream. It might only be a small difference, but I think it worth noting that Schoof's result here isn't supposed to apply to real ice shelves in general, just the SSA setup for which he derived a solution.

Yes, Schoof's setup assumes a 2D shelf that is rapidly sliding (hence with negligible vertical stresses). We clarified the text as follows (p. 7, l. 177): "According to theory (Schoof, 2007a), an unbuttressed, two-dimensional ice shelf in steady state can be melted completely from below…"

**Page 7**

Figure 2. The GL is drawn differently in the left plot compared to the middle and right plots. I think this is probably accidental, but it is confusing to the reader. Please make them look more similar. I note that there seem to be some minor alignment issues with the boxes with each figure, but these are small enough not to be much of a distraction.

We modified the figure as suggested (now Fig. 3).

**Page 8**

Lines 183 – 184. Needs rephrasing. Either this has been demonstrated elsewhere, in which case reference it, or else you actually have considered it and decided it is not worth reporting on in this paper, in which case please change "do not consider" to something like "we carried out preliminary tests (not shown) . . . and found this to yield unrealistic. . .".

Yes, we did consider this scheme in preliminary tests and decided it is not worth reporting in this paper. We rewrote this sentence as (p. 9, l. 205):

"We also considered a 'Full Melt Parameterization' in which the full basal melt rate is applied in partly grounded cells. We found in preliminary tests, however, that this scheme drives unrealistic grounding-line retreat in CISM, as in Seroussi and Morlighem (2018), and we will not consider it further."

Line 185. Maybe "small changes" -> "sub-grid resolution changes" is more specific?

We changed "small changes" to "subgrid changes." (p. 8, l. 201)

Lines 198 – 202. Seems like a slightly odd choice to present conclusions at the start of a results section. I don't have a big problem with this, leave it if you like, it just reads a bit odd.

We reworded this paragraph to leave things more open-ended. The revised text is of the form, "Leguy et al. (2014) demonstrated X for a 1D model. Here, we want to see whether X generalizes to more realistic 3D models." Specifically, the text now reads as follows (now on p. 9, l. 217):

"Leguy et al. (2014) used a one-dimensional model to explore the effect of different basal friction laws on grounding-line migration. They found that a resolution of ~1 km (and, under some circumstances, coarser) is sufficient to accurately represent grounding-line motion if the ice sheet is hydrologically well connected to the ocean, or if a GLP is used for basal shear stress. In contrast, ice sheet models without hydrological connectivity or a GLP (including models that participated in MISMIP3d, as shown in Fig. 5 of Pattyn et al. (2013) require very high resolution (~500 m or finer) to accurately capture grounding-line dynamics. This 1D model was practical for running many experiments at low computational cost, but there was no guarantee that the results would generalize to three-dimensional models. We now use CISM for this purpose.

Line 212. Sorry for not referring back to the MISMIP3D paper (lazy reviewer), but one of the most important aspects of the design here is how this steady state is approached. The GL "stickiness" problem can be characterised by the existence of a region of multiple steady states (which is what leads to irreversibility). So depending on which of these states one starts with, reversibility may or may not be shown. There is more about it here: https://tc.copernicus.org/articles/12/3605/2018/. Please add a line or two about the initial conditions and how steady state is approached.

We expanded the text to read (p. 10, l. 236):
"In the first step, labeled 'Stnd', the model is initialized with a uniform 500-m thick slab of ice and spun-up over 20,000 years with a uniform basal shear stress factor. At the end of the spin-up, the ice sheet has reached a steady state in which the change in grounding line location is less than $10^{-3}$ m a$^{-1}$ and the change in ice thickness at the grounding line is less than $10^{-4}$ m a$^{-1}$."

**Page 9**

Line 222. This 4km seems arbitrary in the absence of any knowledge about the size of the GL response to the perturbation. I presume this is much greater than 4km, so please give some indication of this! Also, knowing the grid cell size would give some context to this. I don't think the reader has yet been given this information at this point? I'm commenting as I read through so I might find this later, but some indication here would be useful to help the reader decide whether 4km is a sensible choice.

We agree that this discussion is premature and that the threshold of 4km is somewhat arbitrary. We repeated the analysis for thresholds of 1 km and 2 km and found that the finding of reversibility is not very sensitive to the chosen threshold, but this needs to be made clear in the text.

We deleted the sentence (p. 10, l. 248), "We will consider an experiment to be reversible if the difference in grounding-line location is 4 km or less." The 4-km criterion is now introduced later, in Sect. 3.3.

On p. 13, l. 300, we added the following text: "Since the 4-km threshold for reversibility is somewhat arbitrary, we note that the results are not very sensitive to this threshold. With a 2-km threshold, the SSA and DIVA tests with p = 0 are labeled as irreversible at 4-km resolution. When the threshold is reduced to 1~km, the BP test with p = 0 at 2-km resolution becomes irreversible. Otherwise, Fig. 5 is unchanged."

**Page 10**

Lines 235-236. If they're directly comparable, why not add Schoof's GL position to the plot? Just as a horizontal line?

We added Schoof's solution to the plot. (now Fig. 4)

**Page 16**

Line 339. "100-m" -> "100m thick " Page 19

Changed to "is initialized as a uniform slab with H = 100 m" (p. 18, l. 399)

**Page 19**

Figures 9 – 12. The different lines are hard to distinguish. I think this is partly because the circles are a bit too large given how close the lines are, and they overlie each other quite often. It is also partly because the connecting line segments are in black and the circles are outlined in black. In Figure 10 especially it is hard to distinguish between 8km and p5km (perhaps because

the circles are smaller but still outlined in black, which dominates?). These figures need to be clearer. It might be that changing the black to the appropriate colour will fix it, but it is hard to say without actually seeing a modified version. I also find "p5" instead of 0.5 a bit counter-intuitive. It looks like something a programmer might write but I don't see why you wouldn't just use 0.5 in a paper. Minor detail: you refer to "vertical line" in Fig 9 caption but "vertical lines" in fig 10 caption. Did you mean to mention also the 100m line again in Fig 10?

Thank you for these suggestions.  We changed the figures as follows (now Fig 10-15):
- removed the black contours around the circles;
- connected the circles with lines of matching color;
- replacing "p5" by "0.5" throughout;
- modified the Fig. 10 caption to explain both vertical lines.

**Page 26**

Line 496. Here you say that results are sensitive to choice of p, but you don't actually state that you get better convergence with p=1. Better convergence with p=1 is expected based on past studies, and I would say your results do indeed support this. So I think you can make a stronger statement here than just saying that the results are sensitive to choice of p.

Our results are not consistent on whether there is better convergence with p = 1.  In the MISMIP3d experiments, convergence is better with p = 1 when running without a GLP, but not with a GLP, as shown in what is now Fig. 4.  In some melt experiments, convergence with p = 1 is significantly slower than with p <= 0.5, as shown for the FCMP and PMP moderate-melt experiments in what is now Fig. 10.

In the statement on now line 608, we agree that "less benefit" is a bit misleading, since in some cases there is no discernible benefit.  We changed the wording to (l. 609) "although MISMIP3d runs show little or no benefit from a GLP when p = 1."

Line 507. Brondex 2018 (J. Glac) is probably the best reference here.

The study by Brondex et al. (2017, J. Glac.) compares Weertman, Schoof/Tsai, and Budd friction laws, but does not explore which value of p is most appropriate for realistic settings such as the Amundsen sector.  At lines 506–507, we cited the study of Joughin et al. (2019) because it links directly to PIG observations.

We agree that Brondex et al. (2017) is relevant to the choice of basal friction law.  We added this study, along with several other references, after the first sentence of this paragraph (p. 31, l. 614): "The choice of basal friction law remains a source of great uncertainty in ice sheet models."

**Page 27**

Line 527. There is a great mystery behind "numerical details of the ice sheet model" which I hope the authors (and the wider ice sheet modelling community) will investigate in the near future. One thought that springs to mind and is not explicitly mentioned here is that the grid or mesh alignment may have some impact. I think it is fairly typical in MISMIP+ simulations for the central section of the grounding line to be aligned approximately across the flow. So if a non-adaptive structured grid is used (which is the case in the current study) then grounding line retreat may naturally occur a row at a time. If previous studies that found NMP to be the better scheme have used an unstructured mesh of triangular elements (ISSM) or an adaptive mesh (BISICLES), it might be worth considering that single element ungrounding may occur more easily in such setups than in the current CISM setup, and perhaps this could explain (part of) the difference? Please don't feel under any obligation to consider this half-baked speculation in the current paper, though perhaps it could be one factor to consider when delving further into this problem.

Yes, the difference in model behavior between ISSM and CISM came as a surprise to us, and we do not yet understand the reason for these differences. We considered diving into the numerics of ISSM and other models to look for explanations, but decided it was better left for a future study. We added a short parenthetical suggestion, changing the text to (p.32, l. 637) "...numerical details of the ice sheet model (including, perhaps, the grid structure and the staggering of variables)." We think your speculation is intriguing and worthy of investigation.

Final paragraph. I am not convinced that this quantification of adequate resolution is fully supported by your simulations. Bear in mind that readers will mostly not read the full paper in detail but will look at key figures and conclusions. This suggestion of 2 - 4km being in general sufficient is dependent on many factors, not all of which are fully explored here, and in particular, this paragraph refers to the MISMIP3D experiments and not to the melt experiments. When I look at Figures 9-12 I do not have confidence that a converged result has been achieved at 2-4 km resolution. If you see, in the next year or two, a published future projection of the Antarctic Ice Sheet carried out at 4km resolution, and read that their justification for the choice of resolution is simply citing your paper, will you feel comfortable with that?

This is a fair point. We do not want to imply blanket approval for runs carried out at 4-km resolution. The final paragraph does mention the melt experiments, and we think that Figs. 9–12 support a resolution requirement of 2–4 km with $p \leq 0.5$. We agree that with $p = 1$, the support for a 4-km grid is weaker. We also acknowledge that this study is far from exhaustive.

In the revised text, we made the last paragraph (now on p. 32) more cautious with the following changes:
- Changed "4 km" to "2–4 km" in the first sentence
- Changed "careful initialization" to "careful initialization and verification" in what was the last sentence. Verification would include the kind of experiments analyzed in this paper.
- Added the following sentence at the end: "Finer resolution of 1--2 km may be needed when using basal sliding laws with Coulomb behavior or strong connectivity to the ocean." This is consistent with the last sentence of the abstract, and would discourage

models with Coulomb sliding laws (including CISM) from using this study as justification for 4-km resolution.

In other recent work (Lipscomb et al. 2021), we ran ISMIP6-style Antarctic simulations at resolutions of 2 km, for comparison to the baseline resolution of 4 km. We found moderately increased sensitivity to ocean forcing with finer resolution, compared with much higher sensitivity to the basal melt parameterization and basal sliding scheme. We therefore added this text to the next-to-last paragraph (p. 32, l. 648): "Lipscomb et al. (2021) found moderate sensitivity to grid resolution in multi-century, ocean-forced Antarctic Ice Sheet experiments with CISM when comparing results at 2 km and 4 km. This sensitivity was less, however, than the sensitivity to sub-shelf melting and basal friction parameterizations."

We are conscious of the fact that most experiments in the Lipscomb et al. (2021) study were run at 4-km resolution, which is probably too coarse for Coulomb laws. We are working at making the model efficient enough for routine use at higher resolution.